# Mechanisms of drug interactions between translation-inhibiting antibiotics

Bor Kavčič [1], Gašper Tkačik [1] & Tobias Bollenbach [2 ✉]

Antibiotics that interfere with translation, when combined, interact in diverse and difficult-to-predict ways. Here, we explain these interactions by "translation bottlenecks": points in the translation cycle where antibiotics block ribosomal progression. To elucidate the underlying mechanisms of drug interactions between translation inhibitors, we generate translation bottlenecks genetically using inducible control of translation factors that regulate well-defined translation cycle steps. These perturbations accurately mimic antibiotic action and drug interactions, supporting that the interplay of different translation bottlenecks causes these interactions. We further show that growth laws, combined with drug uptake and binding kinetics, enable the direct prediction of a large fraction of observed interactions, yet fail to predict suppression. However, varying two translation bottlenecks simultaneously supports that dense traffic of ribosomes and competition for translation factors account for the previously unexplained suppression. These results highlight the importance of "continuous epistasis" in bacterial physiology.

[1] Institute of Science and Technology Austria, Am Campus 1, A-3400 Klosterneuburg, Austria. [2] Institute for Biological Physics, University of Cologne, Zülpicher Str. 77, D-50937 Cologne, Germany. ✉email: t.bollenbach@uni-koeln.de

nhibiting translation is one of the most common antibiotic modes of action, crucial for restraining pathogenic bacteria[1]. Antibiotics targeting translation interfere with either the assembly or the processing of the ribosome, or with the proper utilization of charged tRNAs and translation factors (Fig. 1a, b; Table 1)[2]. Still, the exact modes of action and physiological responses to many such translation inhibitors are unclear. Responses to drug combinations, which may offer effective ways to combat antibiotic resistance[3], are even harder to understand. Apart from their clinical relevance, antibiotic combinations provide powerful quantitative and controlled means of studying perturbations of cell physiology[4]—conceptually similar to studies of epistasis between double gene knockouts[5,6]. Recently, mechanism-independent mathematical approaches to predict the responses to multi-drug combinations were proposed[7,8], yet these approaches rely on prior knowledge of pairwise drug interactions, which are diverse and have notoriously resisted prediction. They include synergism (drug effect is stronger than predicted), antagonism (drug effect is weaker), and suppression (one of the drugs loses potency)[9,10] (Fig. 1c). To design optimized treatments, the ability to predict or alter drug interactions is crucial.

Such predictions would be facilitated by understanding their underlying mechanisms[11].

Predictions of drug interactions should ideally only require information about responses to individual antibiotics. Established null models of drug interactions are based on mechanism-independent expectations such as Loewe additivity (Fig. 1)[12], which mainly serve as a reference for classifying drug interactions. There is currently no null model that captures well-understood processes such as drug uptake, target binding, and the physiological response to target inhibition, which are relevant for all drugs that share the same target. Any deviations from predictions of such a null model could expose drug interactions that cannot be explained by established biological and physical processes alone. Consequently, an improved null model could offer a plausible mechanism for some drug interactions and at the same time expose more complex situations where additional molecular or physiological details are crucial.

Translation is a fundamental, yet complex multi-step process that still lacks a comprehensive quantitative description. A key step toward such a description are bacterial "growth laws," which quantitatively capture the compensatory upregulation of the

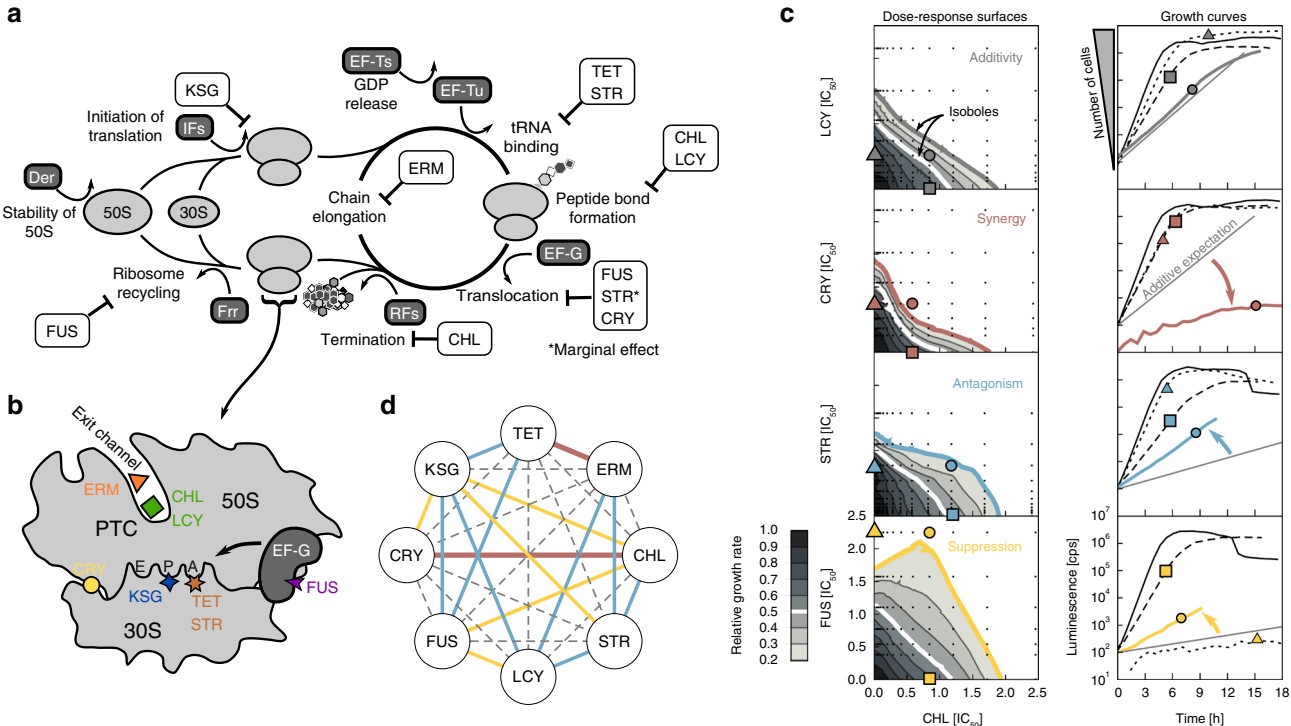

**Fig. 1 Antibiotics targeting different translation steps show diverse drug interactions. a, b** Schematic of the translation cycle and translation inhibitors. Translation factors are shown in dark gray boxes. The stability of the large subunit is mediated by Der and initiation by initiation factors (IFs). Elongation factors Tu and G (EF-Tu, EF-G) catalyze ribosome progression. The release of GDP from EF-Tu is facilitated by EF-Ts. Release factors (RFs) facilitate the ejection of the finished peptide from the ribosome, whose recycling is mediated by the factor for ribosome recycling (Frr). Translation inhibitors are shown in white boxes (abbreviations in Table 1). **c** Examples of response surfaces for different antibiotic combinations corresponding to different interaction types (left column) and examples of growth curves (right column). Left: dose–response surfaces for different drug combinations. Grayscale shows a normalized growth rate as a function of concentrations of two antibiotics. Drug interactions are determined based on the shape of the contour lines of equal growth (isoboles). If the addition of the second drug has the same effect as increasing the concentration of the first, the isoboles are straight lines[12]. Deviations from this additive expectation reveal synergism (the combined effect is stronger and isoboles curve towards the origin), antagonism (the effect is weaker and isoboles curve away from the origin), or suppression (at least one of the drugs loses potency due to the other). Symbols show drug conditions in which growth curves shown in the right column were measured. Right: Growth curves (i.e., time courses of luminescence) for four conditions (no drug, individual drugs, and combination); thin gray line shows the additive expectation of the growth curve for the combined stress; luminescence (photon count-per-second) on y-axes is a proxy for the number of bacteria (Methods). Symbols on the growth curves indicate the condition used: no symbol, triangle, square and a circle correspond to no drug, CHL-only, the second drug only (see left plots), and the combination of both, respectively. The growth curves were shifted in time to originate from the same point at time zero. **d** Drug-interaction network of translation inhibitors. Color-code is as in (**c**); dashed gray lines denote additivity. Each drug interaction was measured twice.

**Table 1 Translation-targeting antibiotics used in this study.**

| Antibiotic | Abbreviation | IC$_{50}$ [μg mL$^{-1}$] | Mode of action, notes |
|---|---|---|---|
| Chloramphenicol | CHL | 1.55 ± 0.06 | Binds in the vicinity of the peptidyl transferase center (PTC) on the 50S subunit[2]; partially overlaps with the aminoacyl moiety of tRNA on the A-site[61]. |
| Lincomycin | LCY | 280 ± 10 | Lincosamide antibiotic; binds next to PTC and interferes with peptide bond formation[2]. |
| Erythromycin | ERM | 25 ± 1 | Macrolide antibiotic that binds further down the nascent peptide exit channel (Fig. 1b), and physically blocks the egress of some newly synthesized peptide chains[2]. Some nascent peptide chains can bypass this block, leading to proteome modification[62,63]. |
| Kasugamycin | KSG | 127 ± 5 | Aminoglycoside; interferes with translation initiation by destabilization of the P-site initiator tRNA and mRNA[64]. |
| Streptomycin | STR | 2.55 ± 0.02 | Aminoglycoside; interferes with the tRNA binding on the A-site and marginally lowers the rate of translocation[2,15,33]. It additionally induces mistranslation[15]. |
| Tetracycline | TET | 0.32 ± 0.01 | Interferes with the binding of aminoacyl-tRNA to the A-site[65]. |
| Capreomycin | CRY | 24 ± 1 | Inhibits translocation by binding to the interface between subunits and stabilization of the ribosome in the pre-translocation state of the ribosome. It only binds the 70S ribosome and not the individual subunits[66]. |
| Fusidic acid | FUS | 64 ± 3 | Inhibits elongation by preventing dissociation of EF-G from the ribosome and lowers the rate of ribosome recycling[67]. |

Antibiotic names, abbreviations, IC$_{50}$ measured for *E. coli* MG1655 in LB medium at 37°C, and notes on their mode of action.

translational machinery in response to perturbations of translation[13]. Growth laws have enabled a model that explains the growth-rate dependent bacterial susceptibility to individual translation inhibitors[14]. Well-defined translation steps cannot only be perturbed chemically[2,15], but also genetically, as these steps are regulated by translation factors—specialized proteins that mediate the stability of ribosomal subunits, catalyze the assembly of 70S ribosomes and initiation, deliver charged tRNAs to the ribosome, release finished peptides, and mediate ribosome recycling (Fig. 1a). Both genetic and chemical perturbations obstruct the progression of ribosomes along the translation cycle, which generally results in a lower growth rate. Comparing the effects of antibiotics to those of precisely defined genetic perturbations offers an opportunity to elucidate the mechanisms responsible for drug interactions between translation inhibitors and could quantitatively test the equivalence of genetic and chemical perturbations of bacterial physiology.

Here, we hypothesize that a key determinant of interactions between pairs of translation inhibitors are the specific steps in the translation cycle where the two inhibitors halt ribosomal progression (Fig. 1a). As a second key determinant of these drug interactions, we consider the compensatory physiological response to translation inhibition captured quantitatively by ribosomal growth laws[13] together with the kinetics of antibiotic transport and ribosome binding. We show that these determinants suffice to explain most drug interactions between translation inhibitors and that these interactions can be predicted solely from known responses to the individual drugs. To establish this result, we use a combination of precise growth measurements, quantitative genetic perturbations of the translation machinery, and theoretical modeling.

## Results

**Diverse drug interactions between translation inhibitors.** To systematically map the network of drug interactions between translation inhibitors, we selected eight representative antibiotics that interfere with different stages of translation and bind to different sites on the ribosome (Fig. 1a, b; Table 1). We determined high-resolution dose–response surfaces for all pairwise combinations of these antibiotics by measuring growth rates in two-dimensional drug concentration matrices using a highly precise technique based on bioluminescence[5,16,17] (Fig. 1c and Supplementary Fig. 1; Methods). The shape of the contour lines, along which growth rate is constant in two-drug space, reveals the

drug interaction type (Fig. 1c). To quantify the drug interactions, we defined the Loewe interaction score LI, which integrates deviations from Loewe additivity (Supplementary Methods). In this way, we characterized all twenty-eight pairwise interactions and constructed the interaction network between the translation inhibitors (Fig. 1d).

The translation inhibitor interaction network (Fig. 1d) we measured has several notable properties. First, antibiotics with similar mode of action tend to exhibit additive drug interactions: In particular, there are purely additive interactions between capreomycin (CRY) and fusidic acid (FUS), which both inhibit translocation, and streptomycin (STR), which interferes with tRNA binding and also slightly lowers the translocation rate. Chloramphenicol (CHL) and lincomycin (LCY), which both inhibit peptide bond formation, interact additively as well. This observation is consistent with the view that drugs with similar mode of action can substitute for one another. Second, kasugamycin (KSG) is a prominent hub in the network: it shows almost exclusively antagonistic and suppressive interactions with other translation inhibitors. Third, we identified a previously unreported synergy between CRY and CHL. Several other interactions confirm previous reports. For example, synergy between erythromycin (ERM) and tetracycline (TET) was observed before[5,18]. Additivity between CHL and TET was also reported; moreover, this interaction proved to be highly robust to genetic perturbations[11]. Finally, antagonism and suppression are more common in the translation inhibitor interaction network than synergy, consistent with a general prevalence of antagonistic interactions between antibiotics[19]. We reasoned that general trends like the prevalence of antagonism in the drug interaction network may be due to a general physiological response to translation inhibition.

**Growth-law based biophysical model does not explain suppression.** As a first step toward understanding the origin of the observed drug interactions, we developed a mathematical model that predicts such interactions from the effects of the individual drugs alone. We generalized a biophysical model for the effect of a single antibiotic on bacterial growth[14] to the situation where two antibiotics are present simultaneously. In the spirit of ref. [14], our model aims to predict the response to a pair of "generic" translation inhibitors whose action leads to a physiological response that obeys established bacterial growth laws[13]. Considerable experimental support from observations of the effects of

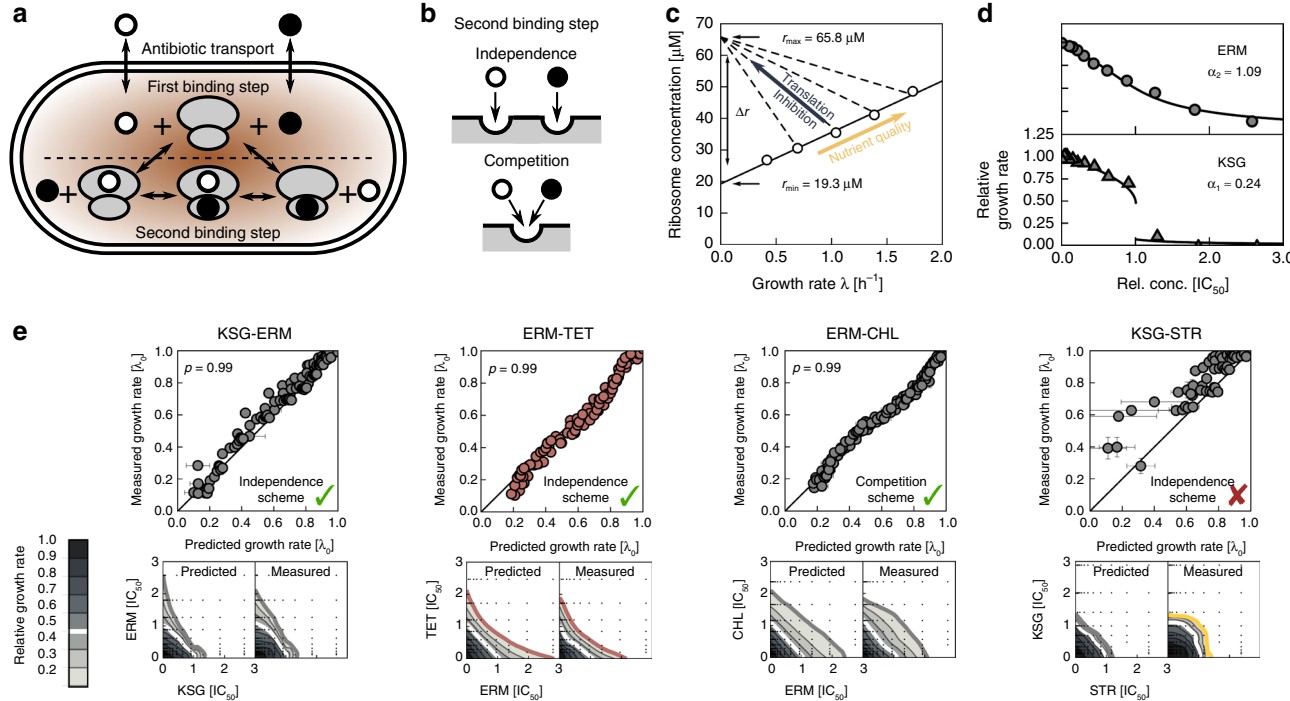

**Fig. 2 Mathematical model partially predicts drug interactions. a** Schematic of antibiotic binding and transport into the cell. Antibiotics (circles) bind to the unbound ribosomes (gray) in the first binding step (above dashed line); bound ribosomes can be bound by a second antibiotic (second binding step; below dashed line). **b** Schematic of antibiotics binding independently (top) or competing for the same binding site (bottom). **c** Growth laws link intracellular ribosome concentration to the growth rate. Solid line: ribosome concentration when the growth rate is varied by varying nutrient quality; dashed lines: ribosome concentration when the growth rate is lowered by perturbation of translation. Circles show data from ref. [14]. **d** Data points are dose–response curves for ERM and KSG; lines show the best-fits of the mathematical model. The best-fit values of the response parameter $\alpha$ that encapsulates kinetic and physiological parameters (Supplementary Information) are shown. Both shallow (top panel, ERM) and steep (bottom panel, KSG) dose–response curves are observed. **e** Examples of predicted dose–response surfaces. The scatter plot depicts the correlation between predicted and measured growth rates. Means and error bars (standard deviation) of predicted growth rates are estimated from $n = 100$ bootstrap repetitions. The binding scheme assumed is indicated on the bottom right and Pearson's $\rho$ on the top left. Predicted and measured dose–response surfaces are shown below the scatter plot. Color of 20% isobole (bottom) denotes the type of predicted interaction. The model correctly predicts response surfaces for KSG-ERM, ERM-TET, and ERM-CHL, yet it fails to predict the interaction between STR and KSG.

diverse chemical and genetic perturbations on translation strengthens the generality of growth laws[20–22]. This parsimonious approach keeps the number of unknown model parameters to a minimum. Cases, where the response to a drug pair deviates from the predictions of this physiologically relevant null model, indicate that more complex mechanisms—specific to one or both of the drugs used—are essential for understanding the drug interaction.

The model consists of ordinary differential equations that capture passive antibiotic transport into the cell, binding to the ribosome (Fig. 2a, b), dilution of all molecular species due to cell growth, and the physiological response of the cell to the perturbation (Fig. 2c). Antibiotic uptake and efflux are characterized by permeability constants. Ribosome binding obeys mass-action kinetics and, in particular, captures the extent to which binding is reversible. The cell's physiological response to translation inhibition is described by bacterial growth laws[13,14], which quantitatively connect the growth rate to the total abundance of ribosomes when the growth rate is varied by the nutrient quality of media or by translation inhibitors. In the model, growth laws determine the regulation of ribosome synthesis in response to translation inhibitors and relate the concentration of free ribosomes to the growth rate. For a single such translation inhibitor, the shape of the dose–response curve depends solely on a response parameter $\alpha$ that merges multiple kinetic parameters into a single number (Supplementary

Information and ref. [14,23]). Notably, all parameters of this model can be inferred from the dose–response curves of individual drugs (Fig. 2d).

When two different antibiotics are present, separate variables are needed to describe ribosomes that are bound by either of the antibiotics individually or simultaneously by both (Fig. 2a, b). Apart from notable exceptions such as the lankamycin-lankacidin and dalfopristin-quinupristin combinations[24,25], the biochemical details of direct physical interactions between antibiotics on the ribosome are largely unknown. Therefore, we assumed that the antibiotic binding and unbinding rates are independent of any previously bound antibiotic; alternatively, to describe competition for the same binding site, we assumed that double-bound ribosomes cannot form (Fig. 2b). These two cases do not require any modifications of the binding parameters inferred from single-drug dose–response curves. The resulting model directly predicts dose–response surfaces rather than fitting them by adjusting free parameters. As a result, this biophysical model provides a well-defined mechanistic null model informed by cell physiology.

Using this model, we calculated the predicted response surfaces for all translation inhibitor pairs and compared them to the experimentally measured surfaces (Supplementary Methods, Fig. 2e, and Supplementary Fig. 2). Certain drug interactions were correctly predicted by this approach (e.g., KSG-ERM, ERM-TET in Fig. 2e), indicating that binding kinetics and growth physiology alone suffice to explain these interactions. Correctly

predicted drug interactions include additive cases which often involve antibiotics that have either similar modes of action (CRY-FUS, CHL-LCY) or partially overlapping binding sites (CHL-LCY, ERM-CHL)[2]. For the latter, the assumption that the formation of the doubly-bound ribosome population is prohibited, which yields an additive response surface, offers even better agreement with the experimental data (Fig. 2e). Occasionally, drug interactions are better explained if competitive binding is assumed (e.g., CHL-TET) even though the binding sites of the antibiotics involved do not overlap.

Other drug interactions clearly deviated from the model predictions. An example is the suppressive/antagonistic interaction between STR and KSG, which was predicted to be additive (Fig. 2e). Such clear deviations could originate from direct molecular interactions of the drugs on the ribosome, and thus be specific for every drug pair; we explore this situation theoretically in ref. [23]. Alternatively, these drug interactions could result from the multi-step structure of the translation cycle itself, which our model does not take into account. Simple partitioning of ribosomes into different populations that are susceptible to different antibiotics does not alter the drug interaction (Supplementary Methods). In the most complex cases, drug interactions could result from drug effects that are unrelated to the primary drug target[11], in particular from effects on drug uptake or efflux[26]. We focused on the plausible hypothesis that drug interactions are caused by the interplay of ribosomes halted in different stages of the translation cycle such as initiation, translocation, recycling, etc. (Fig. 1).

**Inducible translation bottlenecks affect antibiotic efficacy.** To test this hypothesis, we developed a technique for measuring how halting ribosomes in different stages of the translation cycle affects the efficacy of various antibiotics. Specifically, we imposed artificial bottlenecks in translation by genetically limiting the expression of translation factors that catalyze well-defined translation steps[20]. We constructed *E. coli* strains with translation factor genes under inducible control of a synthetic promoter[27]. These genes were integrated into the chromosome outside of their endogenous loci and the endogenous copy of the gene was disrupted (Fig. 3a; Methods). This procedure yielded six strains that enable continuous control of key translation processes (Fig. 3b): stabilization of the 50S subunit (*der*), initiation (*infB*), delivery of charged tRNAs (*tufA/B*), release of GDP from elongation factors (*tsf*), translocation (*fusA*) and recycling of ribosomes (*frr*)[28].

Reducing translation factor expression by varying the inducer concentration resulted in a gradual decrease in growth which stopped at almost complete cessation of growth, reflecting the essentiality of translation factors (Fig. 3c, Methods, and Supplementary Fig. 3a). Since the endogenous regulation of translation factors generally follows that of the translation machinery[29–32], limiting the expression of a single translation factor imposes a highly specific bottleneck as all other components get upregulated. Any global feedback regulation is left intact as we removed the factor from its native operon. Similar genetic perturbations further conform to bacterial growth laws[13,20,21], supporting that translation factor deprivation is a suitable means of assessing responses to targeted perturbations of translation. While antibiotics often have secondary targets and other non-specific effects on the cell, thus obfuscating experiments, translation factor deprivation is highly specific. Our synthetic strains offer precise control over artificial translation bottlenecks that determine the rates of different translation steps and enable disentangling phenomena that are caused by the primary mode of action of antibiotics from those that result from other effects of these drugs.

We used these synthetic strains to assess the impact of bottlenecks on antibiotic efficacy. We measured growth rates over a two-dimensional matrix of concentrations of inducer and antibiotic for each of the six strains (Fig. 3c; Methods and Supplementary Information). To assess if the action of the antibiotic is independent of the translation bottleneck, we analyzed these experiments using a multiplicative null expectation. Note that additivity, as used for antibiotics (Fig. 1c), is not a suitable null expectation here since the responses to increasing concentrations of antibiotic and inducer are opposite. However, if antibiotic action is independent of the translation bottleneck, the growth rate should be a product of the relative growth rates of each of the two perturbations acting individually. Independence implies that the dose–response surface is obtained as a multiplication of the antibiotic dose–response and the translation factor induction curve. Deviations from independence indicate a nontrivial interaction between the bottleneck and the antibiotic action.

We systematically identified interactions between translation inhibitors and bottlenecks by their deviation from independence. In general, antibiotic action can be alleviated or aggravated by a given bottleneck, i.e., the bacteria can be less or more sensitive to the antibiotic due to the bottleneck, respectively. We quantified the magnitude of these effects by bottleneck dependency (BD) scores (Supplementary Methods) and collected them into a single bottleneck dependency vector per antibiotic. The components of this vector describe the interactions between the antibiotic and all six translation bottlenecks. Bottleneck dependency vectors were diverse (Fig. 3d), indicating that bottlenecks at different stages of the translation cycle differentially affect antibiotic efficacy. These results are consistent with the hypothesis that the high diversity of drug interactions between translation inhibitors (Fig. 1d) originates in the diversity of translation steps targeted by the drugs (Fig. 1a).

The bottleneck dependency vector of a given antibiotic provides a quantitative, functional summary of its interactions with the translation cycle. In this sense, it is a characteristic "fingerprint" of the antibiotic. Clustering of antibiotics based on their bottleneck dependency vectors (Supplementary Information) robustly grouped antibiotics with a similar mode of action (CRY and FUS, LCY, and CHL in Fig. 3e, respectively). Notably, this approach separated the translocation inhibitors CRY and FUS from STR, which only weakly affects translocation[33]. Drug interactions between antibiotics from the same cluster were strictly additive (Figs. 1d and 3e). These results show that interactions of antibiotics with translation bottlenecks have explanatory power for drug mode of action and can expose antibiotics acting as substitutes for one another.

While the clustering of certain antibiotics can be rationalized from their presumed modes of action, this is more challenging for others. To further assess the value of this analysis, we measured bottleneck dependencies for three additional antibiotics: lamotrigine (LAM), trimethoprim (TMP), and nitrofurantoin (NIT). As we elaborate in the Supplementary Discussion, using drugs with a defined mode of action (LAM and TMP) corroborates the utility of clustering by bottleneck dependencies, while the similarity of STR to NIT, which has multiple modes of action, suggests a plausible reason for the separation of STR from other clusters of translation inhibitors.

**Effects of translation bottlenecks predict drug interactions.** We reasoned that the effects of translation bottlenecks on antibiotic action should also have predictive power for drug interactions between translation inhibitors. We, therefore, sought a quantitative

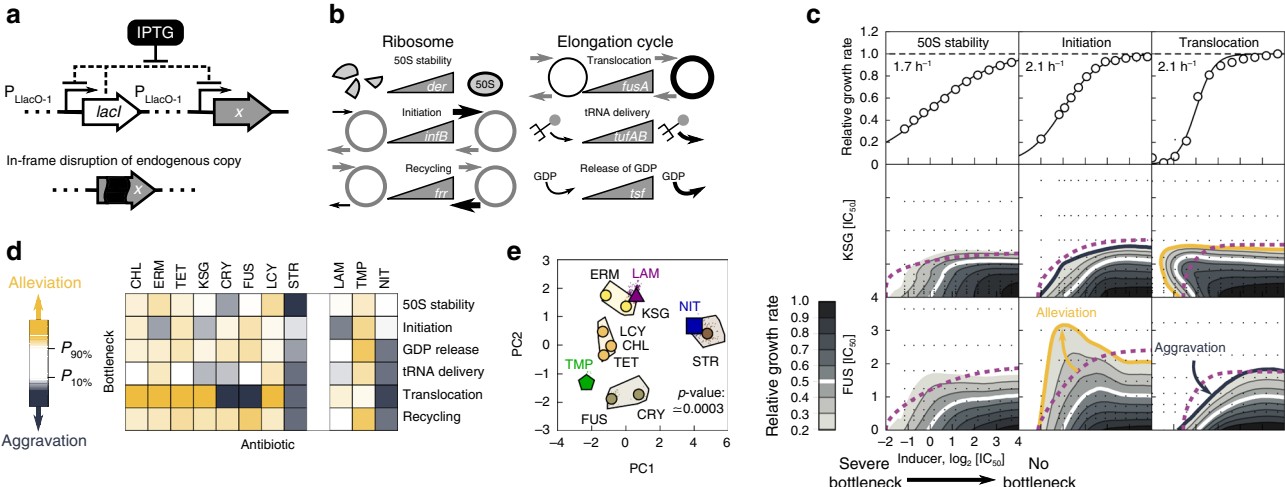

**Fig. 3 Artificial translation bottlenecks strongly affect antibiotic efficacy. a** Schematic of synthetic regulation introduced to control the expression of a translation factor *x*, which creates an artificial bottleneck in translation at a well-defined stage; *lacI* codes for the Lac repressor, which represses the P$_{LlacO-1}$-promoter (Methods[27],). **b** Constructs were made for six translation factors mediating 50S stability (*der*), initiation (*infB*), recycling (*frr*), translocation (*fusA*), tRNA delivery (*tufAB*) and GDP release (*tsf*), respectively. Higher expression alleviates the artificial bottleneck. Thicker lines or arrows indicate higher rates. **c** Translation factor induction curves (upper row) and response surfaces over the inducer-antibiotic grid for different antibiotics (KSG and FUS, middle and bottom row, respectively) in combination with different bottlenecks (50S stability, initiation, and translocation). Full induction of the translation factor rescues wild type growth; increasing bottleneck severity leads to a smooth decrease in growth rate to zero. Induction curves were measured in *n* = 8 technical replicates, and the median value of non-zero growth rates was calculated. Comparison of the response surfaces with independent expectation (dashed purple line) identify alleviation (orange line) or aggravation (dark blue line). **d** Columns show bottleneck dependency vectors in color-code; dependency vectors quantify the response of a given antibiotic to the translation bottlenecks (Supplementary Information). **e** Clustering of the bottleneck dependency vectors upon dimensionality-reduction by principal component analysis (PCA; Supplementary Information). Circles show dependency vectors projected onto the first two principal components (PC1, and PC2); colors indicate cluster identity. The extended cluster areas shown are convex hulls of bootstrapped projections (denoted by dots). Projections of the three additional antibiotics LAM, NIT, and TMP are denoted by a purple triangle, blue square, and green pentagon, respectively. We estimated the *p*-value by clustering *n* = 10[4] reshuffled datasets with added noise and counting the fraction of instances that matched the shown clustering result. See Supplementary Equations (19–20) and Supplementary Fig. 3e; we did not use a standard statistical test.

way of probing the contribution of translation bottlenecks to drug interactions between translation inhibitors.

Ribosomes progress through the translation cycle in a sequence of steps (Fig. 4a). Antibiotics and genetic translation bottlenecks hinder this progression by reducing the transition rates between these steps. If an antibiotic specifically targets a single translation step and reduces the same transition rate as a genetic translation bottleneck, the effects of the drug and the bottleneck should be equivalent, i.e., the consequences of any perturbation elsewhere in the translation cycle should be independent of the exact means by which such a reduction is achieved (Fig. 4b).

To establish the equivalence of specific translation bottlenecks and antibiotic action, we first transformed the measurements of growth rate as a function of translation factor induction into dose–response curves of a corresponding idealized antibiotic that targets a single translation step with perfect specificity. In essence, this procedure converts inducer concentrations into equivalent antibiotic concentrations: the two concentrations are identified as equivalent if they lead to the same relative growth rate (Fig. 4c, d; Supplementary Methods). If the perturbations of factor and antibiotic are equivalent, then the true and the idealized antibiotic should act as substitutes for each other, and exhibit an additive drug interaction. Conversely, we can use this comparison (Fig. 4e and Supplementary Fig. 4) to test systematically if the action of antibiotics is quantitatively equivalent to specific translation bottlenecks.

We found that the effect of certain translation inhibitors is almost perfectly mimicked by translation bottlenecks. Within our selection of antibiotics, several strong candidates for equivalent

perturbations exist (Fig. 1a): CRY, FUS, and potentially STR with EF-G (translocation); KSG with IF2 (initiation); and TET with EF-Tu (tRNA-delivery). For example, remapping the response to CRY and EF-G yields an additive surface (Fig. 4e, f), corroborating that CRY and the EF-G translocation bottleneck are equivalent perturbations. In contrast, if the bottleneck is not equivalent to the drug, remapping does not yield an additive response surface; an example is CRY and the recycling bottleneck (Fig. 4g). Occasionally, marginal effects dominate the apparent equivalence: STR lowers translocation rate only two-fold[33], but inhibiting translocation by deprivation of EF-G is still the best mimic of STR. In general, demonstrating that the action of an antibiotic is equivalent to a specific translation bottleneck provides strong quantitative evidence for its primary mode of action, since translation factors control individual steps with high specificity.

In contrast, the common approach of overexpressing the drug target does not provide useful insights into the mode of action of ribosome-targeting antibiotics. Simple overexpression requires a well-defined drug target like a single protein; overexpressing the ribosome is impractical[34] and would not help distinguish the precise action of different ribosome-targeting antibiotics. Even for less complex drug targets, the interpretation of overexpression assays is challenging[35]. Still, we tested if simple overexpression of translation factors can provide similar insights into the mode of action of TET as translation bottlenecks. Overexpression of translation factors only weakly affected antibiotic efficacy (Supplementary Fig. 8). The effects of overexpressing different translation factors were not specific for antibiotic mode of action

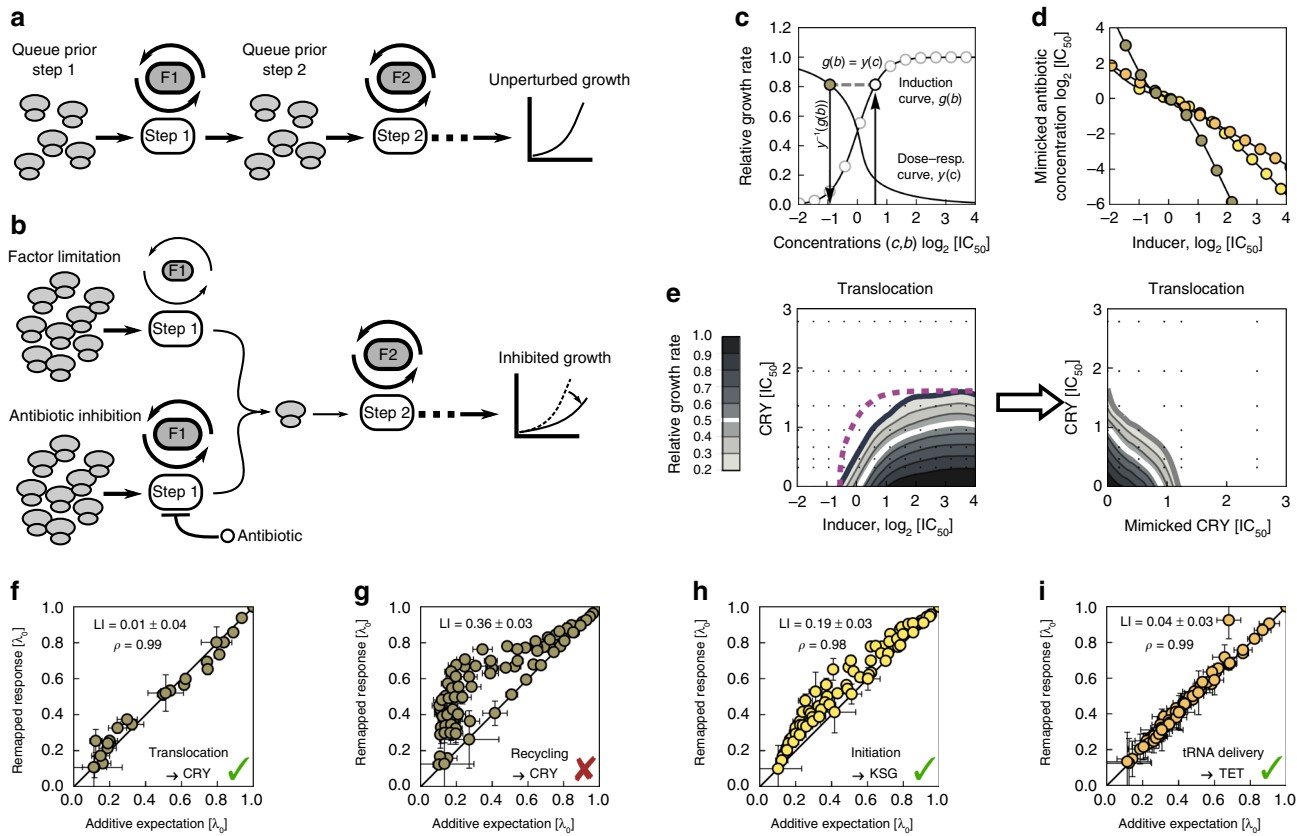

**Fig. 4 Translation factor deprivation mimics the action of equivalent antibiotics. a** Schematic of translation as a sequence of steps (white), catalyzed by translation factors (gray). In the absence of perturbations, ribosomes progress through the steps unimpeded, resulting in unperturbed growth. **b** Schematic of perturbed translation. Top: as the abundance of factor F1 is lowered (smaller factor symbol), the rate of step 1 decreases (thinner arrows) and ribosomes queue in front of the bottleneck. Bottom: the same rate is reduced by an antibiotic. The effects of factor deprivation and antibiotic action on growth are equivalent. **c** Schematic of conversion of inducer concentration $b$ (here for the translocation factor) into the mimicked antibiotic concentration $c$ (here: CRY). For each inducer concentration $b$, the growth rate from the induction curve $g(b)$ is determined and the same growth rate on the antibiotic dose–response curve $y(c)$ is identified (gray dashed line); the inverse function of the dose–response curve yields the equivalent antibiotic concentration as $c = y^{-1}(g(b))$. **d** The resulting conversion of inducer concentration $b$ into antibiotic concentration $c$ for three different pairs of equivalent perturbations: CRY-translocation (gray), KSG-initiation (yellow), and TET-tRNA delivery (orange). **e** Inducer-antibiotic response surface (left) and mimicked antibiotic-antibiotic response surface (right) upon conversion of inducer concentration as in **c**, **d**. Purple dashed line shows isobole for multiplicative responses at relative growth rate 0.2. The remapped response surface is additive, corroborating the equivalence of CRY and translocation factor deprivation. **f–i** Comparison of response surfaces remapped to the additive expectations. The bottlenecks and antibiotics are shown on the bottom right, respectively. Errors in LI and in expected and remapped responses were evaluated by bootstrapping (Supplementary Methods, Supplementary Fig. 4). **f** Additive expectation from **e** and remapped response surface agree ($\rho = 0.99$). **g** As **f**, but for a recycling bottleneck. The large and statistically significant discrepancy in LI from 0 indicates that CRY and a recycling bottleneck are not equivalent. **h** As **f**, but for KSG and an initiation bottleneck ($\rho = 0.98$). **i** As **f**, but for TET and a tRNA delivery bottleneck ($\rho = 0.99$).

(Supplementary Fig. 8). Hence, unlike the depletion of translation factors, their overexpression provides no information about drug interactions with other antibiotics.

For antibiotics that are equivalent to specific translation factors (Fig. 4f–i), drug interactions with other antibiotics can be directly predicted from translation bottleneck measurements. In practice, this is done by remapping the antibiotic-translation factor response surfaces as described above (Fig. 5a–c). Unlike the predictions of the biophysical model (Fig. 2), the predictions made in this way are not based on a mathematical model, but rather on empirical effects of genetic perturbations, which are quantitatively converted into equivalent drug effects; in particular, they are independent of the assumptions underlying the biophysical model. While the biophysical model is only valid for antibiotics that conform to bacterial growth laws, the predictions based on the observed effects of translation bottlenecks are independent of whether or not the growth laws hold for the specific perturbations of translation used. The resulting prediction will be faithful if the

drug interaction originates exclusively from the interplay of two translation bottlenecks.

Drug interactions predicted using this procedure were often highly accurate (Fig. 5). In particular, some of the most striking cases of antagonistic and suppressive interactions were correctly predicted. For example, the prediction of antagonism between CHL and KSG was quantitatively correct (Fig. 5c, d). The same interactions were correctly predicted for LCY (Fig. 5f), which is similar to CHL (Figs. 1a, b and 3e). Remapping qualitatively accounted for nearly all observed interactions of KSG with quantitative agreement in several cases (Supplementary Fig. 5), including the previously unexplained KSG-STR interaction (Fig. 5f). Further, suppression of FUS by CHL was correctly predicted: FUS loses potency when CHL is added (Fig. 5f). In this way, several drug interactions with previously elusive mechanisms are explained by the interplay of the specific steps in the translation cycle that are targeted by the antibiotics involved.

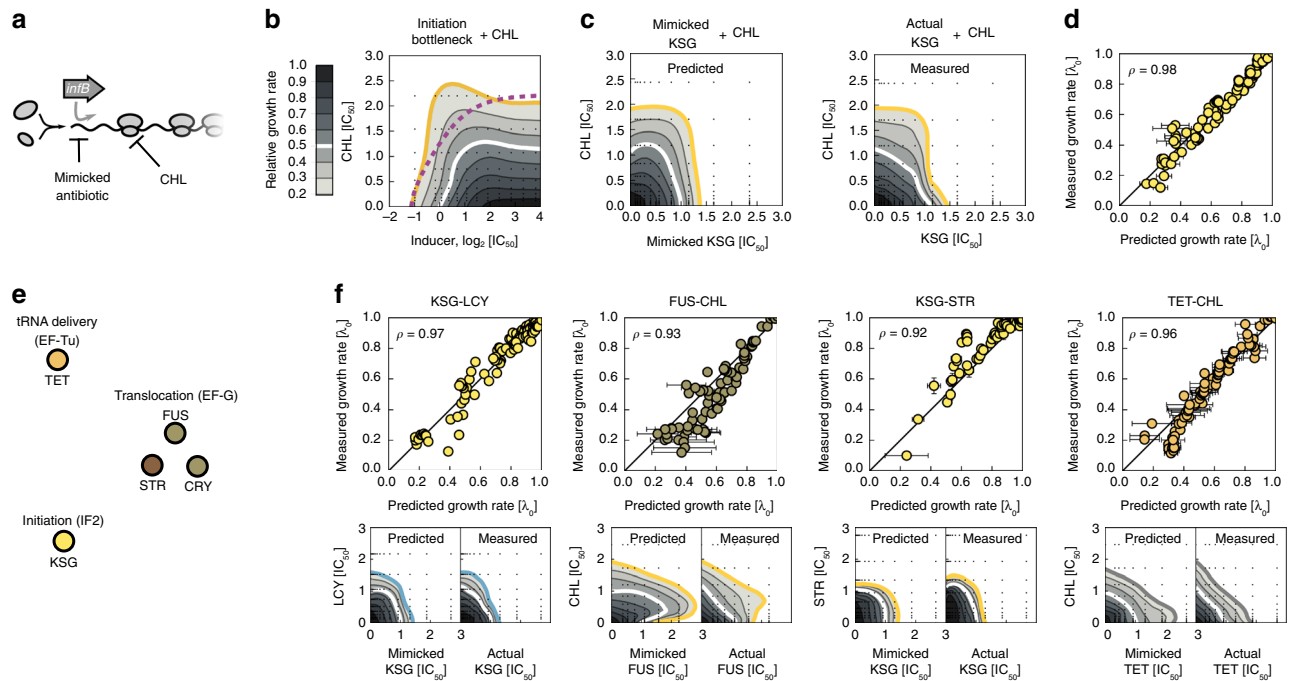

**Fig. 5 Translation bottlenecks can predict antibiotic interactions. a** Example of drug-interaction prediction based on the equivalent translation bottlenecks. The drug interaction between CHL and an antibiotic that targets initiation can be predicted through mimicking the initiation inhibition by limiting the expression of initiation factor (*infB*). **b** The response surface of CHL combined with the inducer for the initiation (*infB*) bottleneck shows mild alleviation. This response surface contains information about the interaction between CHL and any antibiotic that interferes with initiation. The inducer axis is remapped into mimicked antibiotic concentration (Fig. 4c, d). **c** Left: resultant prediction of the response surface for the initiation-inhibiting antibiotic KSG and CHL. Right: measured KSG-CHL response surface for direct comparison; strong antagonism is observed as predicted. **d** A point-by-point comparison of predicted and measured response surfaces (Pearson's $\rho = 0.98$). **e** Schematic showing antibiotics and their equivalent translation factor bottlenecks. Drug interactions with these antibiotics can be predicted for any antibiotic with a known response to the equivalent bottleneck. Color-code shows cluster identity from Fig. 3e. **f** Comparison of predicted and measured response surfaces for different antibiotics in combination with antibiotics that have a factor analog. Top row: scatter plots as in (**d**); bottom row: predicted and measured response surfaces, respectively. Remapping correctly predicts antagonism (KSG-LCY), suppression (FUS-CHL), strong antagonism (KSG-STR), and additivity (TET-CHL).

Remapping correctly predicted additive drug interactions between antibiotics that could not be easily explained by the biophysical model. As noted above, additivity between CHL-TET was predicted for the competitive binding scheme; yet, competitive binding is difficult to rationalize as TET and CHL bind to different subunits. Similarly, the additive pair KSG-ERM is even more puzzling since, unlike ERM, KSG does not act within the elongation cycle (Supplementary Discussion).

For some antibiotic pairs, the predictions based on equivalent translation bottlenecks failed to explain the observed drug interactions (e.g., for LCY-CRY and CHL-CRY; Supplementary Information), indicating that these interactions have origins outside of the translation cycle. We expect that these cases are often due to idiosyncrasies and secondary effects of the drugs, which will require separate in-depth characterization in each case. In contrast, our results show that various non-trivial drug interactions between antibiotics are systematically explained by the interplay of specific translation bottlenecks caused by the antibiotics. While the growth-law based biophysical model already explained ≈57% (16 of 28) of the observed interactions (Supplementary Fig. 2), these included many weak or additive interactions; the most striking suppressive interactions were only captured after taking into account the multi-step nature of translation (Supplementary Fig. 5), thereby increasing the explained fraction to ≈71% (20 of 28).

If suppressive drug interactions are caused by the interplay of different translation bottlenecks alone, it should be possible to recapitulate these interactions in a purely genetic way. We thus

expanded our translation bottleneck approach by introducing multiple genetic bottlenecks in the same cell.

**Double titration of translation factors emulates suppression.** We focused on the interactions between initiation inhibitors (such as KSG) and translocation inhibitors (such as CRY and FUS), which were exclusively antagonistic or suppressive (Fig. 1d). Moreover, the initiation inhibitor KSG alleviated a genetic translocation bottleneck and an initiation bottleneck suppressed the effect of the translocation inhibitor FUS (Fig. 3c). These observations suggest that a universal mechanism underlies the suppression between initiation and translocation inhibitors.

We constructed a synthetic strain that enables simultaneous independent control of initiation and translocation factor levels. We integrated the initiation and translocation factors outside their native loci under the tight control of promoters inducible by isopropyl β-D-1-thiogalactopyranoside (IPTG) and anhydrotetracycline (aTc), respectively, in a strain in which their endogenous copies were deleted (Fig. 6a and Supplementary Fig. 6; Methods). To maximize the precision of induction that is achievable with different inducer concentrations, we put both factors under negative autoregulatory control by chromosomally integrated repressors[13,36]. The resulting strain showed virtually no growth when at least one of the inducers was absent but unrestricted wild type growth in the presence of both inducers (Fig. 6b). These observations confirm that both translation factors are essential and show that their expression can be varied over the

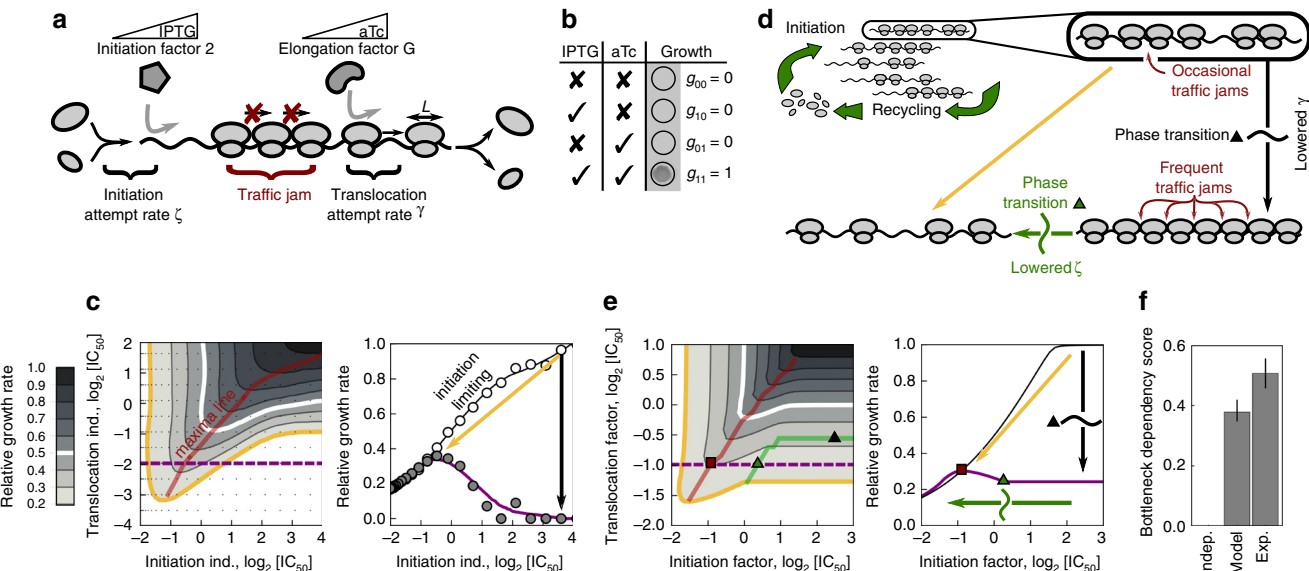

**Fig. 6 Ribosome traffic jams underlie suppression between inhibition of translocation and initiation. a** Schematic of ribosomes progressing along a transcript—a stuck ribosome can cause a traffic jam. Ribosomes undergo factor-mediated initiation events with attempt rate $\zeta$ and translocation with attempt rate $\gamma$. Expression of initiation and elongation factors are controlled by the level of inducer (IPTG and aTc, respectively). **b** Results of all-or-nothing growth assay: bacteria grow only when both essential factors are induced. **c** Left: measured growth rate response surface for the dual inducible promoter strain from (**a**) as a function of both inducer concentrations; the red line shows the ridge of maximum growth. Right: cross-section of the response surface along the dashed purple line (gray circles) and at maximal aTc induction (white circles); solid lines are smoothed profiles. Black arrow denotes a decrease in translocation; if initiation is lowered simultaneously with translocation (orange arrow), growth reduction is smaller. **d** Schematic of the theoretical model: translation is described as an ensemble of transcripts competing for the limited and growth-rate-dependent pool of ribosomes. Ribosomes advance on transcripts as described by a generalized totally asymmetric simple exclusion process (TASEP) for particles of size $L$ (see **a** and text). When $\gamma < \zeta(1 + L^{1/2})$, ribosomes saturate and traffic jams develop, resulting in a drop in elongation and growth (black arrow, the transition happens at the black triangle) (Supplementary Methods[59,60],). When $\zeta < \gamma/(1 + L^{1/2})$, a phase transition occurs (green triangle): traffic jams dissolve—elongation and growth increase (along the green arrow). **e** Left: the growth rate predicted by the generalized TASEP model recapitulates suppression of translocation inhibition by lowered initiation; note that, unlike in (**c**), axes show the concentrations of translation factors. States below and to the right of the green line are in the translocation limiting regime. Right: cross-sections of the response surface. As the initiation factor level is decreased, the critical point of the phase transition (green triangle) is reached; growth starts increasing after passing the critical point, and decreases again after passing the maximum (red square) as the number of translating ribosomes becomes limiting. **f** Bottleneck dependency (BD) score quantifies the deviation from independent expectation (BD = 0) for the response surfaces in (**c**, **e**); heights of bars corresponds to the medians and error bars are 90% bootstrap confidence intervals. Medians and confidence intervals were estimated from $n = 100$ bootstrap data points.

entire physiologically relevant dynamic range, thus enabling quantitative genetic control of two key translation processes.

Curtailing translation initiation suppresses the effect of a genetic translocation bottleneck. We determined the bacterial response to varying translocation and initiation factor levels by measuring growth rates over finely resolved two-dimensional concentration gradients of both inducers. The resulting response surface clearly showed that inhibition of initiation alleviates the effect of translocation inhibition (Fig. 6c and Supplementary Fig. 6). This phenomenon exactly mirrors the antibiotic-antibiotic (KSG-FUS, Fig. 1d) and bottleneck-antibiotic interactions (initiation-FUS, Fig. 3c). An all-or-nothing approach (Fig. 6b), analogous to common genetic epistasis measurements[37], would miss this suppressive effect, highlighting the importance of using quantitatively controlled perturbations. Taken together, these data show that the interplay of translation initiation and translocation alone is sufficient to produce strong suppression: dialing down initiation cranks up growth stalled by translocation bottlenecks. The widespread suppression between antibiotics targeting initiation and translocation is thus explained as a general consequence of the combined inhibition of specific translation steps alone.

What is the underlying mechanism of the suppressive interaction between initiation and translocation inhibitors? We hypothesized that this suppression results from alleviating

ribosome "traffic jams" that occur during translation of transcripts at low translocation rates (Fig. 6d). The traffic of translating ribosomes that move along mRNAs can be dense[38]. When a ribosome gets stuck, e.g., due to a low translocation rate, it blocks the translocation of subsequent ribosomes. The resulting situation is similar to a traffic jam of cars on a road. Traffic jams can form due to the asynchronous movement and stochastic progression of particles in discrete jumps, which is a good approximation for the molecular dynamics of a translating ribosome. If particle progression were deterministic and synchronous, no traffic jams would form. A classic model of queued traffic progression, which can be applied to translation[39,40], is the totally asymmetric simple exclusion process (TASEP)[41,42].

We developed a generalization of the TASEP that describes the traffic of translating ribosomes on mRNAs and takes into account the laws of bacterial cell physiology. There are several differences between the classic TASEP and translating ribosomes moving along a transcript. First, a ribosome does not merely occupy a single site (codon), but rather extends over 25 nucleotides ($\approx 8.33$ codons)[43,44]. Second, the total number of ribosomes in the cell is finite and varies as dictated by bacterial growth laws[13,45]. Third, translation steps are mediated by translation factors that bind to the ribosome in a specific state and push the ribosome into another state[28]. These transitions are stochastic with rates that depend on the abundance of ribosomes in a specific state and on

the abundance of translation factors available to catalyze the step. Thus, the initiation- and translocation-attempt rates, which are constant in the classic TASEP, depend on the state of the system. We formulated a generalized TASEP that captures these extensions, estimated all of its parameters based on literature, and derived the model equations analytically (Methods and Supplementary Information). The resulting growth rate was calculated numerically. In brief, our generalized TASEP model provides a physiologically realistic description of the factor-mediated traffic of ribosomes on multiple transcripts.

Without any free parameters, this generalized TASEP qualitatively reproduced the suppressive effect of lowering the initiation rate under a translocation bottleneck (Fig. 6e). This suppression results from a phase transition between the translocation- and the initiation-limited regime (Supplementary Information). In the translocation-limited regime (black arrow in Fig. 6e), ribosome traffic is dense and cannot be further increased by boosting the initiation attempt rate. Upon decreasing the initiation attempt rate $\zeta$ (green arrow in Fig. 6e), a phase transition to the initiation-limited regime occurs. Beyond the critical point of this phase transition (green triangle in Fig. 6e), the elongation velocity, and with it the growth rate, begins to increase with decreasing initiation attempt rate. Hence, ultimately, a non-equilibrium phase transition in which ribosome traffic jams dissolve underlies the suppressive effect.

The densification of ribosomes on transcripts has an additional consequence: as the number of ribosomes that are stuck on transcripts increases, more elongation factors are sequestered by ribosomes. This in turn reduces the probability that an individual ribosome is bound by a factor—a necessary condition for the ribosome to attempt a translocation step. This situation results in a positive feedback loop in which the reduced translocation attempt rate further amplifies ribosome congestion.

To compare measured and predicted surfaces, which have different axes, we calculated their respective deviation from independence as for the bottleneck dependency score (Fig. 3d and Supplementary Fig. 3). By this measure, the model faithfully captured the clear deviation from the multiplicative expectation (Fig. 6f); the agreement with the experimental data is good, especially considering that the model results are parameter-free and not a fit to the experimental data.

Taken together, these results show that suppressive drug interactions between translation inhibitors are caused by the interplay of two different translation bottlenecks. Close agreement of the experiments with a plausible theoretical model of ribosome traffic, which captures physiological feedback mediated by growth laws, strongly suggests that suppression is caused by ribosome traffic jams. Such traffic jams result from imbalances between translation initiation and translocation; they dissolve in a phase transition that occurs when one of these processes is slowed, leading to an overall acceleration of translation and growth. Stalled ribosomes facilitate the formation of traffic jams by sequestering elongation factors. We conclude that a non-equilibrium phase transition in ribosome traffic is at the heart of suppressive drug interactions between antibiotics targeting translation initiation and translocation.

## Discussion
We established a framework that combines mathematical modeling, high-throughput growth rate measurements, and genetic perturbations to elucidate the underlying mechanisms of drug interactions between antibiotics inhibiting translation. Kinetics of antibiotic-target binding and transport together with "growth laws", i.e., the physiological response to translation inhibition (Fig. 2), form a biophysically realistic baseline model for predicting antibiotic interactions from properties of individual antibiotics alone. This model explained many interactions, but not all, failing specifically for suppressive interactions. Predictions improved by taking into account the step-wise progression of ribosomes through the translation cycle (Figs. 4, 5). This was achieved by mimicking antibiotic perturbations of this progression genetically, which directly identified the contribution of antibiotic-imposed translation bottlenecks to the observed drug interactions. Finally, to explain the origin of suppressive interactions unaccounted for by the biophysical model, we modeled the traffic of translating ribosomes explicitly. Our results show that translocation inhibition can cause ribosomal traffic jams, which dissolve in a non-equilibrium phase transition when initiation is inhibited simultaneously with translocation, thereby restoring growth (Fig. 6). This phase transition explains the suppressive drug interactions between antibiotics targeting initiation and translocation.

Taken together, our framework mechanistically explained 20 out of 28 observed drug interactions (Fig. 1 and Supplementary Figs. 2, 5), as classified based on stringent criteria (Supplementary Information). While 16 out of 28 interactions were already explained by the biophysical model, these include many weak and additive interactions; in contrast, only the translation bottleneck approach correctly predicted some of the strongest interactions and, in particular, suppression. Furthermore, we only classified predictions as correct if the majority of growth rates across the dose–response surface quantitatively matched the prediction. As a result, cases where the predicted and observed drug interaction type agree, are often still classified as false because the agreement is not quantitative. If the same stringent criteria are applied to replicate measurements of drug interactions (Supplementary Fig. 1), only 75% of measurement replicates are classified as faithful predictions. Thus, our conservative estimate of the fraction of explained interactions (71%) is close to the maximum achievable at our measurement precision. Notably, even cases rejected as quantitatively different can provide valuable insights. For example, the remapping-based prediction of the CHL-FUS interaction (Fig. 5f) is rejected because it quantitatively exaggerates the suppression between these drugs. Nevertheless, remapping correctly predicts the occurrence of suppression as well as its direction. Qualitative observations like these still advance our understanding of drug interactions by highlighting drug interaction mechanisms that are distorted by additional effects of unknown origin.

While we focused on translation inhibitors, key elements of our framework can be generalized to drugs with other modes of action. Specifically, when considering a drug that targets a specific process mediated by an essential enzyme, our approach of equating the deprivation of the enzyme with the action of an antibiotic is readily applicable. Our observations also highlight the advantages of factor deprivation compared to simple overexpression: the former produced a quantitative prediction for drug interactions, while no meaningful prediction could be made from overexpression data (Supplementary Fig. 8). The general approach of depleting key accessory proteins is particularly useful for antibiotics targeting multi-component complexes or in cases where the effects of overexpressing the drug target are difficult to interpret[35].

Mimicking the effects of two drugs with controllable genetic perturbations generalizes the concept of genetic epistasis to continuous perturbations. Epistasis studies compare the effects of double gene knockouts to those of single knockouts and identify epistatic interactions—an approach that can reveal functional modules in the cell[6,37,46]. Our results show that continuous genetic perturbations provide valuable additional information on genetic interactions (Fig. 6). Firstly, the direction of epistatic

interactions cannot be extracted from measurements of single and double mutants. Secondly, the quantitative information obtained from such "continuous epistasis" measurements provides more stringent constraints for mathematical models of biological systems. In particular, continuous epistasis data can be powerful for the development of whole-cell models that describe the interplay of different functional modules in the cell. Thirdly, this approach allows including essential genes in epistatic interaction networks even for haploid organisms, which otherwise requires the use of less well-defined hypomorphs. Hence, continuous epistasis measurements augment all-or-nothing genetic perturbations.

Continuous epistasis measurements further enable a deeper understanding of previously mysterious antibiotic resistance mutations. Specifically, translation bottlenecks that alleviate the effect of an antibiotic expose a latent potential for resistance development. Indeed, mutations with effects equivalent to factor-imposed bottlenecks occur under antibiotic selection pressure. For example, resistance to ERM in *E. coli* can be conferred by mutations in proteins of the large ribosomal subunit, that hinder its maturation and lower its stability[47]. Consistent with this observation, our results indicate that the action of ERM is alleviated by lowering the stability of the 50S subunit (Fig. 3d). Mutations in recycling factor were observed in *Pseudomonas aeruginosa* evolved for resistance to the TET derivative tigecycline[48]. The observed alleviation of TET action by a recycling bottleneck (Fig. 3d) offers a mechanistic explanation for the beneficial effects of these mutations. Mutations in other genes predicted based on the effect of translation bottlenecks may be difficult to observe, especially in clinical isolates, due to the associated fitness cost and selection pressure for reverting the mutations in the absence of antibiotic selection. Beyond mutations conferring resistance to individual drugs, consistent or conflicting dependencies of different antibiotics on translation bottlenecks may further indicate the potential for evolving cross-resistance and collateral sensitivity, respectively[49].

Our work also demonstrates the potential of improved null models for drug interactions that are based on generic biophysical and physiological considerations. The number of parameters is minimal and the biophysical model we presented makes parameter-free predictions. This model is readily extended to capture phenomena such as an inactive fraction of ribosomes (Supplementary Information) or physical interactions between antibiotics on the ribosome[23]. Including more detailed mechanisms, e.g., the interplay between different ribosome states that are targeted by different antibiotics, would require additional parameters with unknown values. In essence, such a detailed model and its parameters would have to be fine-tuned for every antibiotic combination. Meaningful predictions would require independent quantitative measurements of multiple kinetic parameters such as the rates of antibiotic binding to the ribosome in different states; for all practical purposes, such a more detailed model would not be predictive. In contrast, the minimal biophysical model we presented provides an improved null expectation for drug interactions. Deviations from this expectation expose drug interactions for which additional details of the antibiotic-ribosome interaction are important. We showed examples of the latter experimentally by halting the ribosome in specific stages of the translation cycle (Fig. 5). Developing a fully parameterized mathematical model of the translation cycle and how it is affected by different antibiotics is a formidable challenge for decades to come.

Crucial to both the minimal biophysical (Fig. 2) and the TASEP-translation model (Fig. 6) is the validity of the growth laws. By experimental validation of such models, we showed that capitalizing on growth laws in theoretical models can offer valuable insights into the interplay of cell physiology and

antibiotic action. Unexplained deviations are good starting points for the identification of situations in which growth laws are violated. This underscores the importance of elucidating such growth laws in other organisms.

In conclusion, we presented a systematic approach for discovering the mechanistic origins of drug interactions between antibiotics targeting translation. As the translation machinery is highly conserved, the interaction mechanisms for drugs targeting specific steps of translation we uncovered may generalize to diverse other organisms. Our approach of mimicking drug effects with continuous genetic perturbations is general and can be extended to antibiotics with other primary targets, other types of drugs, and other organisms. Our quantitative analysis relies on the established correlation between ribosome content and growth rate in varying growth environments[13]. In the long run, extending our combined experimental-theoretical approach to other types of drugs and other biological systems will enhance our understanding of drug modes of action and interaction mechanisms and provide deeper insights into cell physiology.

## Methods

**Bacterial strains.** *Escherichia coli* K-12 MG1655 strain was used as a wild-type (WT) strain. When necessary, the selection on kanamycin was performed at 25 μg mL$^{-1}$ (for post-recombineering selection, see below) or at 50 μg mL$^{-1}$ (for P1 transduction and plasmid selection). A concentration of 100 μg mL$^{-1}$ was used for ampicillin (pCP20, resistance cassette resolution) and spectinomycin (pSIM19, recombineering). The selection for overexpression plasmids was done at 35 μg mL$^{-1}$ of chloramphenicol.

To measure the bioluminescence time traces, pCS-λ bearing the bacterial *luxCDABE* operon driven by the constitutive λ-P$_R$ promoter was transformed into the strains of interest[16]. Selection for the luminescence plasmid was used during the preparation of glycerol stocks (kanamycin 50 μg mL$^{-1}$) but was omitted during the measurements to avoid unknown interactions between the antibiotics used. The plasmid was stably maintained as we observed no significant fitness defect due to pCS-λ and no apparent spontaneous loss of the plasmid as verified by plating on selective and non-selective plates (Supplementary Fig. 7). To this end, we tracked the growth of bacterial cultures in flasks, shaking in a water bath in four conditions. We either actively selected for plasmid maintenance and/or applied antibiotic stress by adding 2 μg mL$^{-1}$ of CHL, which led to ≈50% inhibition. We measured optical density by standard methods (using Hitachi U-5100 cuvette spectrophotometer); after each measurement, we replenished 1 mL of removed medium with fresh, prewarmed medium and corrected the optical density measurements accordingly. After reaching the late exponential phase, we promptly diluted the culture serially and plated equal volumes on both selective and non-selective plates.

The translation factor titration platform was established in strain HG105 (MG1655 Δ*lacIZYA*)[50]. Briefly, endogenous genes encoding for translation factors were first sub-cloned into the pKD13 vector under the control of P$_{LlacO-1}$ promoter with FRT-flanked kanamycin resistance cassette (kan$^R$) and *TrrnB* terminator upstream and downstream of the gene, respectively[13,27,36,51]. The tandem of kan$^R$ and a gene with all regulatory elements was integrated into the chromosome (*galK* locus) using λ-red recombineering (plasmid pSIM19[52]). The kanamycin resistance cassettes here and in the following steps were resolved using yeast FLP resolvase expressed from pCP20[53]. Loss of the resistance cassette and curing of the pCP20 plasmid were checked by streaking on selection agar plates with antibiotics and by junction PCR (for resolution). Following the resolution of kan$^R$, the endogenous factor was inactivated by in-frame deletion: kan$^R$ was integrated into the gene locus and then resolved, which left a 34 residue peptide[51]. We were unable to introduce kan$^R$ directly into the strain with P$_{LlacO-1}$ driven *frr*; therefore, we first performed the deletion in an auxiliary strain MG1655 bearing the ASKA plasmid with *frr*[54] [JW0167(-GFP)], which complemented the chromosomal deletion when IPTG was added. The deletion was possible in the auxiliary strain, yielding MG1655 Δ*frr*:: kanR. We then moved the deletion by generalized P1 transduction[55]. For *tufAB*, we P1-transduced the deletions (Δ*tufA*::kan$^R$ and Δ*tufB*::kan$^R$) sequentially from the respective gene deletion strains from the KEIO collection[56]. All other deletions were performed directly in the strains of interest using λ-red recombineering using pKD13 as a template for the cassette amplification[51]. In the last step, *lacI* driven by the P$_{LlacO-1}$ promoter (yielding growth-rate independent negative autoregulation[13,36]) together with the FRT-flanked kan$^R$ was integrated into the *intS* locus and the resistance cassette was resolved. The allele Δ*intS*::kan$^R$-P$_{LlacO-1}$-*lacI*-*TrrnB* was moved into the strains by generalized P1 transduction. All chromosomal modifications were validated by PCR. The factor titration platform and the repressor operon were Sanger-sequenced at the integration junctions using PCR primers or a primer binding into the kan$^R$ promoter region (which is upstream of the P$_{LlacO-1}$ promoter prior to the resolution). The final genotype for the strains bearing the factor titration platforms is HG105 Δ*galK*::frt-P$_{LlacO-1}$-*x* Δ*x*::frt Δ*intS*::frt-P$_{LlacO-1}$-*lacI*, where *x* denotes the chosen factor. These strains contained

no plasmids and no antibiotic resistance cassettes but had a single copy of a translation factor under inducible control.

To generate the strain with independently regulated initiation and translocation factors, we started with a strain carrying a single $infB$ copy driven by $P_{LlacO-1}$. Then, the negatively autoregulated $tetR$ repressor was integrated into the chromosome, followed by FLP resolvase-mediated resolution of the selection marker. This enabled the integration of $P_{LtetO-1}$-driven $fusA$ into the $intS$ locus; the resolution was followed by the disruption of the endogenous copy of $fusA$. Furthermore, we introduced a negatively autoregulated $lacI$ into the $xylB$ locus. This yielded a marker-less strain with the two essential genes $infB$ and $fusA$ under inducible, negatively autoregulated, and independent control. The final genotype is: HG105 $\Delta galK$::frt-$P_{LlacO-1}$-$infB$ $\Delta infB$::frt $\Delta ycaCD$::frt-$P_{LlacO-1}$-$tetR$ $\Delta intS$::frt-$P_{LtetO-1}$-$fusA$ $\Delta fusA$::frt $\Delta xylB$::frt-$P_{LlacO-1}$-$lacI$. Oligonucleotide sequences, targeted template, restrictions sites (when used), and a brief description of use are listed in Supplementary Data 1. All DNA modifying enzymes and Q5 polymerase used in PCR were from New England Biolabs.

We constructed overexpression strains by transforming HG105 with pCS-λ and plasmids from the ASKA library[54] and its derivatives. We used ASKA plasmids in which GFP has been excised from the reading frame; we had to repeat the excision of GFP from the $infB$-bearing plasmid by NotI digestion and subsequent ligation as per ref. [54]. All plasmids were Sanger-sequenced. For controls we used (i) plasmid pAA31 (gift from A. Angermayr), in which the open reading frame is cleanly deleted, as transcription-only control, and (ii) the ASKA plasmid bearing $lacZ$ as a neutral protein overexpression control. We note that the overexpression of proteins leads to growth inhibition[13,57]; hence, we actively selected for plasmid maintenance by adding 35 μg mL$^{-1}$ of chloramphenicol into the growth medium.

**Growth rate assay and two-dimensional concentration matrices**. Rich lysogeny broth (LB) medium, which at 37 °C supports a growth rate of $2.0 \pm 0.1$ h$^{-1}$, was used. LB medium was prepared from Sigma Aldrich LB broth powder (L3022), pH-adjusted to 7.0 by adding NaOH or HCl, and autoclaved. Antibiotic stock solutions were prepared from powder stocks (for catalog numbers, see Supplementary Table 1), dissolved either in ethanol (CHL, ERM, and TET), DMSO (LAM and TMP) or water (KAN, CRY, LCY, KSG, FUS, and STR), 0.22 μm filter-sterilized and kept at −20 °C in the dark until used. Antibiotics were purchased from Sigma Aldrich or AvaChem. Some of the antibiotics (e.g., ERM, FUS, and LCY) are not used in the clinic against certain Gram-negative bacteria due to generally poor efficacy; however, at higher concentrations (yet still well below the solubility limit) inhibition of growth is observed.

A previously established growth-rate assay based on photon counting was used to precisely quantify the absolute growth rates for 5–9 generations[16]. Cultures were grown in 150 μL of media in opaque white 96-well microtiter plates (Nunc 236105), which were tightly sealed by transparent adhesive foils (Perkin-Elmer 6050185 TopSeal-A PLUS) to prevent contamination and evaporation. We prepared glycerol stocks of WT and factor-titration platform strains from saturated overnight cultures. We inoculated the cultures with ~$10^2$ cells per well (1:$10^6$ dilution) from either thawed glycerol stocks (for the drug interaction network) or from liquid cultures in which we first incubated the bacteria containing the factor titration-platform for 1 h in the absence of IPTG (inoculated by 1:2000 dilution of the glycerol stock) to partially dilute out the remaining factor molecules before additional 1:1000 dilution into measurement plates. Between 10 and 20 plates were cycled through a plate reader using a stacking system (Tecan M1000, controlled by Tecan i-control software, v1.10.4). We built a custom incubator box around the stacker towers to facilitate ventilation and fix the temperature to 37 °C (Supplementary Fig. 7). This incubator was designed and troubleshot by B.K. and Andreas Angermayr (IST Austria and University of Cologne) and built by IST Miba Machine Shop. Each plate was read every 20–40 min and was shaken (orbital 10 s, 582 rpm) immediately before reading (settle time 10 ms, integration time 1 s). Plates were manually pipetted and concentration gradients of antibiotics and inducers (IPTG, aTc) were prepared by serial dilution (0.70-fold).

Growth rates were determined as a best-fit slope of a linear function fitted to the log -transformed photon counts per second. The fitting procedure and examples of growth curves are shown in Supplementary Fig. 1. In rare cases of occurrence of mutants (as evidenced by sudden growth) we manually removed the measurement (only in the case of $tufA$ titration). We verified that the luminescence-based technique leads to the same results as a classical optical density-based one (Supplementary Fig. 7).

**Normalization of dose–response surfaces**. All growth rates were normalized relative to the average growth rate in the drug-free medium [for factor-titration strains at the highest inducer concentration (5 mM)]. Small differences between individual dose–response curves were inevitable due to known challenges of preparing identical concentrations gradients on different days. To correct for such day-to-day variability, we rescaled the concentration units to the IC$_{50}$ for each drug. The IC$_{50}$ was obtained from fitting the Hill function $y(x) = 1/[1 + (x/\text{IC}_{50})^n]$ to the individual dose–response curves. The dose–response curve of each drug was measured seven times and averaged. The IC$_{50}$ and corresponding errors reported in Table 1 are extracted from such average dose–response curves (Supplementary Fig. 1g). Induction curves were normalized slightly differently, using a shifted and increasing Hill

function in the form $g(b) = [(b + b_0)/\text{IC}_{50}]^n/\{1 + [(b + b_0)/\text{IC}_{50}]^n\}$, where $b_0$ is a concentration offset. The latter parameter was required as the complete cessation of growth was not achievable in some cases even in the absence of inducer as the promoter $P_{LlacO-1}$ is leaky. Inducer concentrations were thus rescaled via $b \rightarrow (b + b_0)/\text{IC}_{50}$.

**Smoothing of dose–response surfaces**. To reduce noise when plotting response surfaces, we smoothed the data using a custom Mathematica script that implements locally weighted regression (LOESS)[58]. This approach only smoothed the contours and did not alter the character of dose–response surfaces. Smoothing was only used for plotting and not for the analysis in which only interpolation between adjacent points was used (Mathematica function Interpolation).

**Statistics and reproducibility**. We measured the dose–response surfaces for all 28 drug interactions in duplicate. As the dose–response surface was measured over a $12 \times 16$ grid, the duplicates swap the drug axes ($12 \times 16 \rightarrow 16 \times 12$ across two 96-well plates) on different days to check for effects coming from spreading the measurements over different plates. The experimental and analysis procedure led to reproducible measurements of growth rates between days (Supplementary Fig. 1, $\rho \approx 0.86$). For the double factor titration experiment, the inducer gradients were set up across 6 plates to form a $24 \times 24$ grid. Each response surface is thus based on multiple measurements and the impact of individual points is assessed by bootstrapping. In total, we measured over 20,000 growth curves. We automated the collection and analysis of the data to allow for unbiased interpretation of the data.

We characterized the type of drug interaction by calculation of the Loewe interaction score [Supplementary Equation (1)]. The effects of bottlenecks on the efficacy of antibiotics were quantified by calculating the bottleneck dependency score [Supplementary Equation (2)]. To classify the interactions between antibiotics and antibiotic-bottleneck pairs, we estimated the null distributions by bootstrapping (Supplementary Information).

We assessed the accuracy of our predictions by "isobole sliding," which measures the average deviation of measured isoboles from predicted ones (Supplementary Information). We performed bootstrapping to statistically determine the significances of predictions.

**A biophysical model for a pair of antibiotics**. We constructed a minimal mathematical model describing the combined antibiotic action based on growth laws[13] and kinetics of antibiotic transport and binding[14]. The corresponding system of coupled ordinary differential equations (ODEs) and additional analysis are specified in the Supplementary Information. Differential equations describe the time-evolution of unbound, individually-bound and double-bound ribosomes, as well as the intracellular concentrations of antibiotics. Growth laws convert the concentration of unbound ribosomes into the growth rate, which determines the total abundance of ribosomes[13,14]. Steady-state solutions were found numerically. A more detailed analysis of model extensions is reported in ref. [23].

**TASEP model of translation**. We developed a mean-field mathematical model of factor-mediated translation which recovers the suppression between inhibitions of initiation and translocation. We took into account that ribosomes can perform a specific step only when bound by a corresponding translation factor. The mathematical framework is detailed in Supplementary Methods followed by a Supplementary Discussion of the effect of mRNA growth-rate dependence, rescue mechanisms, and inefficiency of a direct response to translocation inhibition.

**Materials availability**. Strains described in this study can be obtained from the corresponding author upon reasonable request.

**Reporting summary**. Further information on research design is available in the Nature Research Reporting Summary linked to this article.

## Data availability
Data to produce all figures are available in Source Data, oligonucleotides are listed in Supplementary Data 1. Source data are provided with this paper. Minimally processed raw data is publicly available at https://doi.org/10.15479/AT:ISTA:8097 together with analysis scripts. Source data are provided with this paper.

## Code availability
The scripts for the mathematical models and for the data analysis were implemented in Mathematica 11.3 (Wolfram Research). The scripts have been deposited to IST DataRep and are publicly available at https://doi.org/10.15479/AT:ISTA:8097. Source data are provided with this paper.

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

## Acknowledgements

We thank M. Hennessey-Wesen, I. Tomanek, K. Jain, A. Staron, K. Tomasek, M. Scott, K.C. Huang, and Z. Gitai for reading the manuscript and constructive comments. B.K. is indebted to C. Guet for additional guidance and generous support, which rendered this work possible. B.K. thanks all members of Guet group for many helpful discussions and sharing of resources. B.K. additionally acknowledges the tremendous support from A. Angermayr and K. Mitosch with experimental work. We further thank E. Brown for helpful comments regarding lamotrigine, and A. Buskirk for valuable suggestions regarding the ribosome footprint size. This work was supported in part by Austrian Science Fund (FWF) standalone grants P 27201-B22 (to T.B.) and P 28844 (to G.T.), HFSP program Grant RGP0042/2013 (to T.B.), German Research Foundation (DFG) standalone grant BO 3502/2-1 (to T.B.), and German Research Foundation (DFG) Collaborative Research Centre (SFB) 1310 (to T.B.). Open access funding provided by Projekt DEAL.

## Author contributions

Conceptualization, methodology, formal analysis, investigation, writing—original draft, writing—review & editing: B.K., G.T., and T.B. Funding acquisition, resources, and supervision: G.T. and T.B.

## Competing interests

The authors declare no competing interests.
