## [Peer Review File · Nature Communications]

Reviewers' comments:

Reviewer #1 (Remarks to the Author):

Building upon their expertise in quantitative characterization of antibiotic interactions, the authors present a quantitative characterization, and theoretical explanation, for interactions of pair-wise combinations of ribosome targeting antibiotics in *E. coli*. The manuscript is comprised of three main projects: the high-resolution antibiotic interaction surface (Fig 1), the high-resolution antibiotic-mutant library interaction surface (Fig 3) and the biophysical/mathematical modeling of the phenomenon (Figs 2/6).

The high resolution surface of the antibiotic interactions is a specialty of the Bollenbach lab. As usual, the data is highly reproducible and done under balanced growth conditions. In contrast to past work from this lab, what is unusual about the interaction surfaces is that antibiotics within the same putative class are compared. A naïve assumption would be that only additive interactions would be possible for within-class antibiotics. That is certainly not the case. The authors suggest this is due partly to the interplay between antibiotic action and the physiology of the cell, with the shared physiology necessarily imposing an indirect interaction among the antibiotics, and partly due to the multi-step nature of protein translation.

To validate the mechanistic mode of action, the antibiotics are compared with strains that have the mediators of the various steps of translation (initiation, elongation, ribosome recycling, etc) under the control of inducible promoters. The authors demonstrate that these genetic mutants behave as antibiotic mimics, which in turn suggests that ribosomal traffic jams are the origin of the interactions not explained by the physiological model.

The traffic jam model, based upon the Totally-Asymmetric-Simple-Exclusion-Process (TASEP) toy model developed in physics, has a long history of application in protein translation modeling and ribosomal traffic jams. What is new here is that the host physiology is included. The authors' model represents a substantial step forward in biophysical modeling. Beyond model development, their queuing model suggests the origin of the interactions not captured by physiological coupling alone. As the authors say (Lines 349-351), the model predictions are surprising (reassuringly) good given that there are no free parameters with which to fit the data.

The authors make clear that there is nothing in their approach that is idiosyncratic to ribosome-targeting antibiotics and *E. coli*. A similar experimental framework could be used to validate/discover mode-of-action for any collection of growth inhibitors. The main obstacle is the development of a biophysical model for the interaction between host physiology and the growth inhibitors, but these generalizations are well-underway by other groups.

In short, the work is outstanding, both the experimental and theoretical components. My suggestions are largely cosmetic.

1. I was unable to find the Loewe interaction index scores for the antibiotic combinations that underlie Fig 1D. It would be useful to have those shown, for example, in the upper-right of the contour plots in Supplemental Figure 1. That would provide a simple, scalar gauge of the interaction strengths.

Later in the main text (Lines 267-280), the authors refer to the explanatory power of the biophysical model and the TASEP model. Supplemental Figure 2E conveys qualitative information about which interactions can be explained by the individual models, and which cannot. It would be useful to the reader if there was a quantitative scalar comparison as well. For example, a series of histograms (with error bars) or a box-and-whisker plot comparing the experimental Loewe indices and the Loewe indices predicted by the models. The vertical axis would look like Supplemental

Figure 1E rotated clock-wise a quarter-turn. Or, alternatively, a plot like Fig 2E, but for all the 28 antibiotic combinations compared with the biophysical and TASEP models where appropriate.

Such a plot could then be referred to in Lines 377-387 where the authors distinguish between quantitative and qualitative predictions of the models. It would then be clearer to the reader that the prediction-success-rate of 20-out-of-28 is a conservative underestimate, and that allowing qualitative fits would raise that fraction.

2. In Table 1, one of the outcomes of the biophysical model is that the IC50 is a growth-state-dependent measure (eg. Supplemental eq. (7)). The title (or the IC50 column header) should include the physiological qualifier '...for E. coli MG1655 grown in LB medium at 37C' or similar.

Reviewer #2 (Remarks to the Author):

Kavčič et al. employ mathematics modeling to examine, explain and predict interactions between several types of protein synthesis inhibitors. They further generated specific translation bottlenecks by limiting the production of specific auxiliary proteins (translation factors or assembly chaperones) and showed that limiting the amount of one or another auxiliary factor can mimic the antibiotic action. Their model explains some of the interactions, but not the others.

Critique: Although generally an interesting study, this work suffers from the lack of clarity, inaccurate assumptions and unclear implications. Some of the revelations seem more or less obvious, whereas the others are really hard to understand. The importance of this work for a broad audience is rather unclear.

Specific criticism:

Line 77: Streptomycin is NOT an inhibitor of translocation.

Lines 93-102. The model authors consider seem to be too simplistic because it does not take into account that antibiotic binding may depend on the state of the ribosome. For example, it is unclear how the model accounts for the fact that kasugamycin and possibly tetracycline (see below) may interact only with the initiating, but not elongating ribosome, or how the incompatibility of erythromycin with a longer (but not shorter) nascent protein chain in the exit tunnel would change it.

Lines 177-184. While I agree with authors that some of their data seem to converge nicely, the other are more difficult to explain. For example, why would erythromycin, a known elongation inhibitor would be grouped with kasugamycin, an initiation inhibitor?

Lines 209-218. To this reviewer, it seems almost an obvious assumption that inhibiting a specific step of the translation cycle by an antibiotic would be equivalent to reducing the concentration of an accessory protein that is needed for the corresponding step in translation and thus, should yield similar interaction patterns with the other inhibitors. Therefore, the importance of this section was lost on me. The authors write "In general, demonstrating that an antibiotic acts as an equivalent perturbation to a specific translation factor provides strong evidence for its primary mode of action, since translation factors are thought to control individual steps with high specificity." This seems to be long known – the modes of action of many antibiotics have been determined by observing partial protection of the cell when the suspected target was overexpressed.

Lines 255-262. The authors indicate that in their measurements tetracycline and chloramphenicol exhibited additive interaction because in their view both drugs inhibit tRNA delivery. However, this assumption is likely either incorrect (in regards to TET) or imprecise (in regards to CHL). Recent data showed that tetracycline acts primarily as initiation inhibitor (Nakahigashi et al., 2016, DNA Res. 23, 193-201), whereas chloramphenicol interferes with tRNA delivery only at specific codons (Marks et al., 2016, PNAS, 113, 12150-12151; Choi et al. 2019, Nat Chem Biol, PMID: 31844301).

Other issues:

Lines 105-105 "In the absence of knowledge about direct molecular interactions on the ribosome (as for the pairs of lankamycin and lankacidin or of dalfopristin and quinupristin...". The interactions of streptogramins (dalfopristin and quinupristin) or lankamycin and lankacidin with the ribosome have been extensively studied. Why 'in the absence of knowledge'?

Line 327. When talking about the ribosome extending 'over 16 codons', the authors refer to the 35-old paper. Maybe they would be better of referring to the more modern data coming from ribosome profiling which showed that the ribosome occupies 25-28 nucleotides (e.g. Mohammad et al. (2109) eLife 42591).

Line 433 What is "stacker-based setup for high-throughput growth rate measurements"?

Table 1.

Chloramphenicol: it does NOT overlap with the acceptor stem of tRNA but with its aminoacyl moiety (Dunkle et al., 2010, PNAS, 107, 17152)

Erythromycin: The drug does not apparently "block the egress of the newly synthesized peptide chain". See for example Kannan et al. (2012) Cell 151, 508 or Vazquez-Laslop 2018, TIBS 43, 668.

Kasugamycin: interferes with initiation by destabilization of P-site tRNA AND mRNA.

Capreomycin: by saying 'fully assembled ribosome' do authors mean 70S ribosome vs individual subunits? If so, it is a wrong term. The correct term would be 'only 70S ribosome, not individual subunits'.

Fusidic acid: The drug acts UPON completion of translocation. Therefore, a more accurate term would be 'inhibits elongation by preventing dissociation of EF-G from the ribosome'.

Supplementary information, II.19-20. The growth rate measurements were based on following luminescence mediated by a low copy-number plasmid pCS-lambda. I wonder how it was determined that in the absence of selection there was "no apparent spontaneous loss of the plasmid"

Reviewer #3 (Remarks to the Author):

In this paper, these authors try to predict the interesting combined effect of different translation-targeting drugs. Two major predictive conclusions are used, in brief: growth law based model and translational bottleneck based model. First, I appreciate and agree on the conclusion that the suppressive interaction between initiator inhibitors and translocation inhibitor is due to alleviated traffic jam since it is clear that drugs like Cm causes ribosome stalling across the mRNA, which may cause serious issues of accumulation of stalled ribosomes (although this is intuitively not surprising). I also appreciate the development of a series of artificial translational perturbation systems, which is useful in studying translation.

However, there are important issues that could seriously affect the solidness of the paper.

Major Comments:

The paper is hard to read. The authors omit lots of important content in the main text, which I think is important. The authors should try their best to make the paper be easily accessible to its potential audiences. Figure 1C is clearly a very key data, however, the authors go on without an even detailed explanation of it when writing the result part (or in figure caption). From a first glimpse, I don't even understand how to read the Figure 1C (what's the meaning of the axis of the plot? and what's the several lines in the right column of Figure 1C IC50 plot?). Again for Figure 2, where is the core model (growth-law based bi-physical model) in the main text? Key content should not be put in supplementary file. Although the details may be very long, you should describe some key points in main text. Otherwise, it looks that you are keeping claiming something without any basis.

The most important issue 1: The growth-law model used in the paper is well documented in Greulich 2015 MSB paper. However, there is a serious issue here. Growth-law model is an excellent model in predicting the coordination between gene expression and cell growth. However, it should be cautious when applying growth law to different conditions since it is a phenomenological model that could originate from different molecular origins. For example, the growth-law data in Figure 2B is based solely on chloramphenicol in Scott 2010. Actually, if you try other antibiotics in your list, you might get different R_{max} (try to measure it under Ery, KSG and Tet conditions). Another even more important issue is that: under antibiotic treatment, a large fraction of ribosomes is inactive due to the problem of inhibition of ribosome assembly. So R_{max} , R_{min} in your model is problematic. If you assume drug does not bind $R_{inactive}$, you may get a complete different result. You should do an assessment of how the change of R_{max} , R_{min} could affect your results. Otherwise, the model itself is not solid. As a result, you find some drug-pairs observe your mode while others do not in Figure 2. You should consider my above comment.

The most important issue 2: translational bottleneck model. The authors attempt to use growth-law model to explain a part of drug interactions (57%). Translational bottleneck model doesn't do much further (71%). On one hand, the author should link those two theories (I don't see the relation here). Do those artificial translational perturbation systems also observe the growth-law model? On the other hand, there are many uncertain things regarding using artificial translation perturbations to mimic the antibiotics. Since as the authors also point out, artificial titration of certain translational proteins is certainly a more pure way while a lot of antibiotics have secondary effects which make the situation be more complicated. Therefore, with the imperfectness of both the two models and those uncertainties in the model, the conclusion of the paper is not that convincing.

Minor comments:

To my knowledge, E. coli K-12 strain is not sensitive to fusidic acid due to the poor permeability (please clarify this issue). Therefore, I am not sure how the authors perform fusidic acid experiments. Moreover, for the growth rate measurement, I have a hard time to find an exact concentration of drug that used. Most data is an ambiguous IC50. This is unclear. I suggest the author to show a dose-dependence growth rate data for each drug with (exact concentrations of drug, instead of merely IC50), to standardize the result.

The method of growth rate measurement is very weird to a microbial physiologist. Why does the author adopt such a method? The conventional OD measurement is very easy; you can simply do it with a micro-plate reader.

Point-by-point response

Reviewer #1:

Building upon their expertise in quantitative characterization of antibiotic interactions, the authors present a quantitative characterization, and theoretical explanation, for interactions of pair-wise combinations of ribosome targeting antibiotics in E. coli. The manuscript is comprised of three main projects: the high-resolution antibiotic interaction surface (Fig 1), the high-resolution antibiotic-mutant library interaction surface (Fig 3) and the biophysical/mathematical modeling of the phenomenon (Figs 2/6).

The high resolution surface of the antibiotic interactions is a specialty of the Bollenbach lab. As usual, the data is highly reproducible and done under balanced growth conditions. In contrast to past work from this lab, what is unusual about the interaction surfaces is that antibiotics within the same putative class are compared. A naïve assumption would be that only additive interactions would be possible for within-class antibiotics. That is certainly not the case. The authors suggest this is due partly to the interplay between antibiotic action and the physiology of the cell, with the shared physiology necessarily imposing an indirect interaction among the antibiotics, and partly due to the multi-step nature of protein translation.

To validate the mechanistic mode of action, the antibiotics are compared with strains that have the mediators of the various steps of translation (initiation, elongation, ribosome recycling, etc) under the control of inducible promoters. The authors demonstrate that these genetic mutants behave as antibiotic mimics, which in turn suggests that ribosomal traffic jams are the origin of the interactions not explained by the physiological model.

The traffic jam model, based upon the Totally-Asymmetric-Simple-Exclusion-Process (TASEP) toy model developed in physics, has a long history of application in protein translation modeling and ribosomal traffic jams. What is new here is that the host physiology is included. The authors' model represents a substantial step forward in biophysical modeling. Beyond model development, their queuing model suggests the origin of the interactions not captured by physiological coupling alone. As the authors say (Lines 349-351), the model predictions are surprising (reassuringly) good given that there are no free parameters with which to fit the data.

The authors make clear that there is nothing in their approach that is idiosyncratic to ribosome-targeting antibiotics and E. coli. A similar experimental framework could be used to validate/discover mode-of-action for any collection of growth inhibitors. The main obstacle is the development of a biophysical model for the interaction between host physiology and the growth inhibitors, but these generalizations are well-underway by other groups.

In short, the work is outstanding, both the experimental and theoretical components. My suggestions are largely cosmetic.

We thank the reviewer for her/his careful assessment of our work, the positive comments, and the constructive suggestions for improvements.

1. I was unable to find the Loewe interaction index scores for the antibiotic combinations that underlie Fig 1D. It would be useful to have those shown, for example, in the upper-right of the contour plots in Supplemental Figure 1. That would provide a simple, scalar gauge of the interaction strengths.

We have added the Loewe interaction scores in Supplementary Fig. 1 as suggested by the reviewer. We further added a scatterplot of LI scores from replicate measurements (Supplementary Fig. 1f) to provide a measure of reproducibility and calculated the correlation coefficient between replicates, which offers an empirical upper limit for the correlation between predicted and measured LI scores.

Later in the main text (Lines 267-280), the authors refer to the explanatory power of the biophysical model and the TASEP model. Supplemental Figure 2E conveys qualitative information about which interactions can be explained by the individual models, and which cannot. It would be useful to the reader if there was a quantitative scalar comparison as well. For example, a series of histograms (with error bars) or a box-and-whisker plot comparing the experimental Loewe indices and the Loewe indices predicted by the models. The vertical axis would look like Supplemental Figure 1E rotated clock-wise a quarter-turn. Or, alternatively, a plot like Fig 2E, but for all the 28 antibiotic combinations compared with the biophysical and TASEP models where appropriate.

We agree that a quantitative comparison of the predictions of the biophysical model and the translation factor titration approach with the experimental data is helpful. We have added such a comparison in Supplementary Figs. 2 (new panel f) and 5 (new panel b). We added scatterplots of the predicted and observed (bootstrapped) LI scores rather than including plots for each individual interaction (as in Fig. 2e). Here, we would like to stress that LI is a robust measure of interaction strength (as pointed out by the reviewer) but problematic for evaluating the explanatory power of models: LI integrates the information of the whole surface into a single number, thus masking intricacies of surface shape, see Supplementary Information, lines 279-283.

Such a plot could then be referred to in Lines 377-387 where the authors distinguish between quantitative and qualitative predictions of the models. It would then be clearer to the reader that the prediction-success-rate of 20-out-of-28 is a conservative underestimate, and that allowing qualitative fits would raise that fraction.

We have revised this paragraph of the Discussion (lines 418-434 of the revised manuscript): It now refers to the new comparison in Supplementary Figs. 2 and 5 and clarifies that qualitative agreement between our prediction and the experimental data is achieved for a larger fraction of drug pairs.

2. In Table 1, one of the outcomes of the biophysical model is that the IC50 is a growth-state-dependent measure (eg. Supplemental eq. (7)). The title (or the IC50 column header) should include the physiological qualifier ‘...for E. coli MG1655 grown in LB medium at 37C’ or similar.

We have added this qualifier to the title of Table 1.

Reviewer #2:

Kavčič et al. employ mathematics modeling to examine, explain and predict interactions between several types of protein synthesis inhibitors. They further generated specific translation bottlenecks by limiting the production of specific auxiliary proteins (translation factors or assembly chaperones) and showed that limiting the amount of one or another auxiliary factor can mimic the antibiotic action. Their model explains some of the interactions, but not the others.

Critique: Although generally an interesting study, this work suffers from the lack of clarity, inaccurate assumptions and unclear implications. Some of the revelations seem more or less obvious, whereas the others are really hard to understand. The importance of this work for a broad audience is rather unclear.

We appreciate the reviewer's thorough assessment of our work and the helpful constructive suggestions. It has become clear that several aspects of our work were not explained clearly enough. We strongly revised the manuscript to make it clearer and, in particular, added several explanations that are

more thorough. We also corrected a number of erroneous claims, some of which were due to ambiguous reports in the literature. Further, we toned down and moved more speculative claims that are only tangentially relevant for our main results from the main text into a new Supplementary Discussion. We want to thank the reviewer particularly for pointing out these specific issues along with relevant references: This certainly helped us to improve the manuscript. As explained in more detail below, we further performed additional experiments, which strengthen our key findings. We also added clearer explanations of how our work is relevant beyond improving null models and suggesting plausible mechanisms of drug interactions.

Line 77: Streptomycin is NOT an inhibitor of translocation.

We thank the reviewer for pointing this out. This mistake occurred because we relied on two review articles: Blanchard, S.C., Cooperman, B.S., & Wilson, D. *Chem. Biol* 17 (2010), and Wilson, D. *Nat. Rev. Microbiol.* 12 (2014). The former explicitly states that streptomycin is a translocation inhibitor and Table 1 in the latter confirmed this notion. The reference cited for this [Peske, F. *et al. J. Mol. Biol.* 343 (2004)] reports the decrease in translocation rate to be two-fold, which is indeed quite low compared to other antibiotics interfering with translocation. Yet, Peske *et al.* find that tRNA on the A-site is significantly stabilized (~45-fold decrease in Kd). They suggest that this apparent paradox can be explained by a streptomycin-induced conformational change in the 30S subunit, which is prone to accelerate translocation that in turn compensates for over-stabilization. In the revised manuscript, we note the effect of STR on translocation as marginal, and highlight its dominant mode of action (interference with the binding of tRNA); we have corrected this in Table 1 and in the main text (lines 214-215, 255-257); we also modified Fig. 1A accordingly.

Irrespective of this controversy in the literature, note that our translation bottleneck analysis revealed that the action of STR is equivalent to translocation inhibition (Supplementary Fig. 4). The biochemical data from the references mentioned above might offer an explanation why the principal component projection separates STR from FUS and CRY (Fig. 3e), which are translocation inhibitors. We now briefly mention this point in the Results part where we describe Fig. 3e (lines 224-225) as well as in the Supplementary Discussion (line 459).

Lines 93-102. The model authors consider seem to be too simplistic because it does not take into account that antibiotic binding may depend on the state of the ribosome. For example, it is unclear how the model accounts for the fact that kasugamycin and possibly tetracycline (see below) may interact only with the initiating, but not elongating ribosome, or how the incompatibility of erythromycin with a longer (but not shorter) nascent protein chain in the exit tunnel would change it.

The biophysical model serves first and foremost as an improved null model – a benchmark we use to probe to what extent the drug interactions can be explained by a model that takes key cell physiology into account (via growth laws) but uses only information readily accessible by individual dose-response curve measurements. As such, we can use the same model for all antibiotic pairs, including poorly characterized ribosome-targeting antibiotics. It is correct that the biophysical model does not include details such as the state of the ribosome. We left out these details on purpose since our goal was to develop a minimal model of antibiotic-ribosome interactions in the same spirit as *e.g.*, [Greulich, P. *et al. Mol. Syst. Biol.* 11 (2015)]. This is not to say that these details are not important – on the contrary: For some antibiotics, such details are crucial for understanding the causes of drug interactions. However, a model taking into account all these details would become very complicated and would need to be fine-tuned for each specific antibiotic. Importantly, this would require parameter values that are currently unknown (such as the forward and reverse binding rates of antibiotics to ribosomes in different states)

and, even worse, additional assumptions would have to be made when such antibiotics are combined. As a result, such an extended model would no longer be predictive.

By keeping the model simple, we can make parameter-free predictions, which we compare to the experimental data (Fig. 2). One can think of this as a minimal model that makes predictions for generic translation inhibitors that differ in their ribosomal binding site and kinetics, but for which we do not have any more detailed information. Cases where model predictions fail indicate that the details of how the antibiotics affect the ribosome are important. We address more complex cases later in our manuscript by investigating the effects of translation bottlenecks using an experimental approach (Figs. 3-6).

In the revised manuscript, we now mention this point in the Results part (lines 109-116) and explain it more thoroughly in the Introduction (lines 38-49) and in the Discussion (lines 472-487). In particular, we now comment that modeling the whole translation cycle as well as all antibiotic idiosyncrasies would require a prohibitive number of (predominantly) unknown parameters (lines 476-482).

Lines 177-184. While I agree with authors that some of their data seem to converge nicely, the other are more difficult to explain. For example, why would erythromycin, a known elongation inhibitor would be grouped with kasugamycin, an initiation inhibitor?

We appreciate the reviewer's comment. We find this particular example interesting: It suggests a non-trivial similarity between antibiotics, namely that erythromycin, which cannot block synthesis of (some) proteins above a certain length, may effectively act as an initiation inhibitor. This is an example where our quantitative approach of titrating the expression of translation factors provides novel insights into subtleties of the mode of action of these antibiotics.

We revised the corresponding part of the main text (lines 297-298) and discuss this noteworthy example in more detail in a new Supplementary Discussion (lines 481-488). We emphasize that this similarity in mode of action of erythromycin and kasugamycin remains to be corroborated by other experimental approaches in future work.

Lines 209-218. To this reviewer, it seems almost an obvious assumption that inhibiting a specific step of the translation cycle by an antibiotic would be equivalent to reducing the concentration of an accessory protein that is needed for the corresponding step in translation and thus, should yield similar interaction patterns with the other inhibitors. Therefore, the importance of this section was lost on me. The authors write "In general, demonstrating that an antibiotic acts as an equivalent perturbation to a specific translation factor provides strong evidence for its primary mode of action, since translation factors are thought to control individual steps with high specificity." This seems to be long known – the modes of action of many antibiotics have been determined by observing partial protection of the cell when the suspected target was overexpressed.

We agree that this hypothesis would be straightforward, but only if the precise molecular mode of action of the antibiotic is known and any side-effects (e.g., binding to secondary targets) can be ruled out; however, the former is often, and the latter almost universally, not the case. Hence, in our view, the hypothesis that reducing the concentration of the accessory protein is equivalent to antibiotic action cannot be taken for granted and must be tested. It is correct that overexpressing the drug target is a common way to identify the mode of action of drugs and this works for some, but by far not for all, antibiotics [Palmer, A.C., & Kishony, R. *Nat. Commun.* 5 (2014)]. The translation factors are not the drug targets here, making the interpretation of simple overexpression experiments even more challenging. If this result is known, we would be happy to cite the corresponding references; however, we are not

aware of prior work using this approach and would need the corresponding references to cite them. Importantly, our assessment of equivalence between factor deprivation and antibiotic action is highly quantitative and as such enables a precise classification of equivalent interventions: By continuously varying factor expression, we identify if the depletion of specific translation factors can quantitatively mimic the effect of the antibiotic. Establishing a similarly stringent criterion for overexpression experiments is problematic since quantitative equivalence cannot be achieved in such experiments, which can at best detect a qualitative decrease in efficacy of the antibiotic.

Still, we agree that the advantages of our approach are clarified by a comparison of our results to a simple overexpression of translation factors. To address this point, we performed new experiments where we overexpress the translation factors using a more conventional approach. In general, when the target of the drug is a single enzyme, overexpression offers a way of assessing the action of the antibiotic. However, even in this case, the interpretation of target overexpression is challenging as it can render cells more or less sensitive to the drug, depending on details [Palmer, A.C., & Kishony, R. *Nat. Commun.* 5 (2014)]. The situation is worse for antibiotics targeting a complex structure such as the ribosome with numerous accessory proteins – here it is not even clear what to overexpress: Overexpressing the ribosome is not feasible [Jinks-Robertson, S., Gourse, R.L., & Nomura, M. *Cell* 33 (1983)] and, even if it were, it would not be helpful for distinguishing the modes of action of different ribosome-targeting drugs. The most promising alternative is to overexpress the translation factors. To this end, we have constructed bacterial strains carrying plasmids from the ASKA library [Kitagawa, M., *et al. DNA research* 12 (2005)], which enabled us to perform such overexpression experiments.

We focused on the effects of translation factor overexpression on the action of tetracycline. We measured the growth rate over two-dimensional concentration gradients of inducer and tetracycline. We observed that the relative sensitivity to tetracycline is largely independent of the translation factor that is overexpressed. Here, one might have expected the bacteria to tolerate higher concentrations of tetracycline when we overexpress EF-Tu (delivery of charged tRNAs) or initiation factor 2 (as per the reviewer's comment regarding potential inhibition of initiation by tetracycline) – yet, we detected no such effect. This is in stark contrast to the controlled depletion of translation factors, the effects of which are highly specific for different antibiotics (Fig. 3d). Unlike translation factor depletion, overexpression provides little information about the interaction with other antibiotics. This example illustrates that the controlled depletion of this subset of genes is more informative than their overexpression.

We now mention this comparison to simple overexpression assays in the Results part (lines 260-270), revisit it in the Discussion (lines 438-443), and we added Supplementary Figure 8 showing these results. The technical details of the new overexpression assays are explained in the Supplementary Information (lines 67-74).

Lines 255-262. The authors indicate that in their measurements tetracycline and chloramphenicol exhibited additive interaction because in their view both drugs inhibit tRNA delivery. However, this assumption is likely either incorrect (in regards to TET) or imprecise (in regards to CHL). Recent data showed that tetracycline acts primarily as initiation inhibitor (Nakahigashi et al., 2016, DNA Res. 23, 193-201), whereas chloramphenicol interferes with tRNA delivery only at specific codons (Marks et al., 2016, PNAS, 113, 12150-12151; Choi et al. 2019, Nat Chem Biol, PMID: 31844301).

We agree that this point was not well explained in our original manuscript. This is a relatively minor result of our work, and we moved it into the Supplementary Discussion to avoid a possible distraction from our main results. Following the reviewer's suggestion, we also toned down our interpretation of

these data and explained the context better in lines 475-480 of the Supplementary Discussion; we thank the reviewer for pointing out these references, which we now cite to clarify that the detailed mode of action of these drugs is more complicated.

Other issues:

Lines 105-105 “In the absence of knowledge about direct molecular interactions on the ribosome (as for the pairs of lankamycin and lankacidin or of dalfopristin and quinupristin...”. The interactions of streptogramins (dalfopristin and quinupristin) or lankamycin and lankacidin with the ribosome have been extensively studied. Why ‘in the absence of knowledge’?

We thank the reviewer for pointing this out. Indeed, the wording we used here was misleading. We meant to say that apart from lankamycin-lankacidin or dalfopristin-quinupristin, there are not many other well-understood cases. We rephrased this sentence accordingly (lines 131-134).

Line 327. When talking about the ribosome extending ‘over 16 codons’, the authors refer to the 35-old paper. Maybe they would be better of referring to the more modern data coming from ribosome profiling which showed that the ribosome occupies 25-28 nucleotides (e.g. Mohammad et al. (2109) eLife 42591).

We agree that modern data can offer a more refined view of such parameters. We contacted Allen Buskirk [corresponding author of Mohammad F., Green, R., & Buskirk, A. *eLife* 8:e42591 (2019)] for additional input on this topic. He recommended to use 24-25 nt for the size of the ribosome footprint as this is the distance between collided ribosomes they observed [Woolstenhulme, C.J., et al. *Cell Reports* 11 (2015)]. Therefore, we now use $L=25/3 \approx 8.33$ codons as the ribosome footprint. We also refined the estimate of the mRNA lifetime based on Yu, J., et al. *Science* 311 (2006). These changes do not affect any of the main conclusions previously described in the section “Simultaneous titration of translation factors reveals robust suppression between translocation and initiation inhibition.” We recalculated the dose-response surfaces in Fig. 6e and evaluated the bottleneck dependency scores shown in Fig. 6f. We analyzed the effect of different parameters on the resulting suppression (Supplementary Methods lines 416-429 and Supplementary Fig. 6).

Line 433 What is “stacker-based setup for high-throughput growth rate measurements”?

The stacker-based setup is a plate reader capable of sensitive luminescence measurements over time on many 96-well plates in parallel, located in a custom incubator box to control temperature. We included a detailed schematic of this setup in Supplementary Fig. 7d and added a brief description of the implementation details in the figure caption.

Table 1.

Chloramphenicol: it does NOT overlap with the acceptor stem of tRNA but with its aminoacyl moiety (Dunkle et al., 2010, PNAS, 107, 17152)

Thanks. This has been corrected.

Erythromycin: The drug does not apparently “block the egress of the newly synthesized peptide chain”. See for example Kannan et al. (2012) Cell 151, 508 or Vazquez-Laslop 2018, TIBS 43, 668.

We have expanded this description to include a comment about the selectively-permissive synthesis of proteins and proteome modification caused by erythromycin.

Kasugamycin: interferes with initiation by destabilization of P-site tRNA AND mRNA.

This has been corrected.

Capreomycin: by saying ‘fully assembled ribosome’ do authors mean 70S ribosome vs individual subunits? If so, it is a wrong term. The correct term would be ‘only 70S ribosome, not individual subunits’.

Indeed, this is what we mean; we have corrected it.

Fusidic acid: The drug acts UPON completion of translocation. Therefore, a more accurate term would be ‘inhibits elongation by preventing dissociation of EF-G from the ribosome’.

This has been corrected.

Supplementary information, ll.19-20. The growth rate measurements were based on following luminescence mediated by a low copy-number plasmid pCS-lambda. I wonder how it was determined that in the absence of selection there was “no apparent spontaneous loss of the plasmid”

We agree that it is helpful to explain these controls. In the new Supplementary Fig. 7 we now show data from control experiments we performed, supporting that there is no apparent plasmid loss in the absence of selection. We verified this by plating on selective and non-selective agar plates, in the presence and absence of antibiotic stress, respectively. Following a comment by reviewer #3, we also validated that the growth rates measured using luminescence are identical to those measured using optical density (also shown in the new Supplementary Fig. 7).

Reviewer #3:

In this paper, these authors try to predict the interesting combined effect of different translation-targeting drugs. Two major predictive conclusions are used, in brief: growth law based model and translational bottleneck based model. First, I appreciate and agree on the conclusion that the suppressive interaction between initiator inhibitors and translocation inhibitor is due to alleviated traffic jam since it is clear that drugs like Cm causes ribosome stalling across the mRNA, which may cause serious issues of accumulation of stalled ribosomes (although this is intuitively not surprising). I also appreciate the development of a series of artificial translational perturbation systems, which is useful in studying translation.

We highly appreciate the reviewer's positive comments on the conclusions of the traffic-jam model and on the construction of artificial bottlenecks in translation. We agree that the occurrence of traffic jams is plausible. However, while seemingly intuitive, this effect is not clear *a priori*: The model results show that this effect only occurs in a certain parameter regime; further, the sequestration of elongation factors by stalled ribosomes contributes to this effect, which may be less intuitive. We revised the corresponding part of the Results (lines 381-386) to clarify this point.

However, there are important issues that could seriously affect the solidness of the paper.

Major Comments:

The paper is hard to read. The authors omit lots of important content in the main text, which I think is important. The authors should try their best to make the paper be easily accessible to its potential audiences.

We thank the reviewer for pointing this out and we agree that certain points were not clear enough in the previous version of the manuscript. We have now added several additional explanations to elaborate

on the results in our article. Following this comment (and those of the other reviewers), we strongly revised our manuscript to make it easier to follow and added numerous clarifications throughout the text. We also implemented additional feedback on the presentation, which we received from colleagues.

Figure 1C is clearly a very key data, however, the authors go on without an even detailed explanation of it when writing the result part (or in figure caption). From a first glimpse, I don't even understand how to read the Figure 1C (what's the meaning of the axis of the plot? and what's the several lines in the right column of Figure 1C IC50 plot?).

Thank you for highlighting these specific unclear points, which was very helpful for making these results more accessible. We added additional explanatory legends and labels, in particular a grayscale bar that explains the lines in Fig. 1c and revised the caption for clarity; we also revised the corresponding paragraph in the Results part (lines 82-86) accordingly.

Again for Figure 2, where is the core model (growth-law based bi-physical model) in the main text? Key content should not be put in supplementary file. Although the details may be very long, you should describe some key points in main text. Otherwise, it looks that you are keeping claiming something without any basis.

We agree and have revised the part of the main text where the biophysical model is first introduced (lines 109-129, as well as Methods 524-531), in particular by adding a more thorough description of the key phenomena captured by this model and its conceptual basis. We are not sure if the reviewer had this in mind, but we left the model equations in the supplement since going through these in the main text would become overly technical and would be distracting for most readers. In addition, we modified Fig. 2a,b to define the key processes described by the model more clearly.

In the present manuscript, we focus on our most interesting results, which can be directly compared to experiments. A more technical article focused on a comprehensive theoretical analysis of the biophysical model and several extensions of this model will be published elsewhere (a preprint is available on the bioRxiv: 10.1101/2020.04.18.047886).

The most important issue 1: The growth-law model used in the paper is well documented in Greulich 2015 MSB paper. However, there is a serious issue here. Growth-law model is an excellent model in predicting the coordination between gene expression and cell growth. However, it should be cautious when applying growth law to different conditions since it is a phenomenological model that could originate from different molecular origins. For example, the growth-law data in Figure 2B is based solely on chloramphenicol in Scott 2010. Actually, if you try other antibiotics in your list, you might get different R_{max} (try to measure it under Ery, KSG and Tet conditions). Another even more important issue is that: under antibiotic treatment, a large fraction of ribosomes is inactive due to the problem of inhibition of ribosome assembly. So R_{max} , R_{min} in your model is problematic. If you assume drug does not bind $R_{inactive}$, you may get a complete different result. You should do an assessment of how the change of R_{max} , R_{min} could affect your results. Otherwise, the model itself is not solid. As a result, you find some drug-pairs observe your mode while others do not in Figure 2. You should consider my above comment.

Generality of growth law: We have included additional support for the broad validity of the growth laws. The main data shown in [Scott, M., *et al. Science* 330 (2010)] is indeed based on experiments with chloramphenicol. However, in the supplementary material of Scott *et al.*, several independent confirmations of the observed relation are shown: New experiments with tetracycline and neomycin, as well as literature data for titration of initiation factor 2 [Cole, J.R., *et al. J. Mol. Bio.* 198 (1987)] and 3 [Olsson, C.L., *et al. Mol. Gen. Genet.* 250(1996)]. Moreover, a study [Bennet, P.M., & Maaløe, O. *J. Mol.*

Bio. 90 (1974)] found the same relation for fusidic acid. While additional measurements for all different antibiotics could further strengthen the validity of the growth laws (or expose exceptions), that is beyond the scope of our study and should, in fact, be a topic of a dedicated study on the validity of growth laws. In contrast, in the present work, we assume the validity of the growth laws to make theoretical predictions for drug interactions; this assumption is justified by the considerable experimental support in the references mentioned above.

The purpose of our model is to capture the effects of typical (or ‘generic’) translation inhibitors. In the absence of hard data disputing the growth laws, the only reasonable way to build a model is to assume that these laws hold. From a theory perspective, the whole point of such laws is that they do not need to be re-measured for every situation they are applied to, but rather enable real predictions about outcomes of experiments that are independent of those used in the construction of the model. This is the same rationale that underlies the modeling approach in Greulich *et al.* and other recent papers [e.g., Deris, J.B., *et al.*, *Science* 342 (2013)]. However, we agree with the reviewer that specific antibiotics (or other perturbations of translation) may lead to violations of the growth laws (just as physical laws like Ohm’s law break down outside of their regime of validity). For such antibiotics, our model for generic translation inhibitors will likely make wrong predictions. Exposing these cases, where our current theoretical understanding is insufficient to explain the experimental observations, is one of the key goals of our approach.

To make these points clearer, we included the references mentioned above and now explain the evidence for the generality of the growth laws (lines 109-113) and the possibility that they may be violated for some antibiotics in the Discussion (lines 488-493). We clarify that we do not claim that this model is always valid but rather that, despite its simplicity, it makes surprisingly accurate parameter-free predictions in many cases.

Inactive ribosomes and other extensions of the biophysical model: We thank the reviewer for raising the interesting point about inactive ribosomes. Our model can actually be generalized to capture this phenomenon. Specifically, the effects of different subpopulations of ribosomes (e.g., initiated and non-initiated) that can be bound by different antibiotics can be analyzed analytically. We can show that, as long as ribosomes interchange rapidly between the subpopulations, this effectively only leads to a rescaling of the binding constants, which does not affect the rest of our analysis. We have included this additional analysis in the Supplementary Information (lines 194-206) and now also mention this point in the Results part (lines 474-475).

Describing more details such as multiple different ribosome states, each with different (unknown) binding kinetics for different antibiotics, or (unknown) changes in R_{min} or R_{max} for different antibiotics is feasible in principle, but would result in a much more complicated model with many unknown parameters. In an upcoming more theoretical publication, we derive models of antibiotics with altered growth laws (bioRxiv: 10.1101/2020.04.18.047886). Comparing them to our experimental data would, however, require fitting many parameters, which (i) would not be fully constrained by our data, (ii) would need to be fine-tuned separately for each antibiotic or, even worse, (iii) would need to be fitted separately for each antibiotic pair. Even re-measuring the growth law (and how it might change) for additional antibiotics would not be helpful for making predictions about the drug interactions we focus on in this work: If e.g., R_{min} or R_{max} changed for one antibiotic, we would have to make additional assumptions about how these values change when that antibiotic is combined with another antibiotic. In essence, following this approach would replace our relatively simple model that makes parameter-free predictions with one that is not predictive but can fit almost any data.

We are not saying that details revealed by fitting such models would be unimportant – on the contrary: For some antibiotics, such details will be crucial for understanding the causes of drug interactions. Rather, we argue that our biophysical model, which makes predictions for generic translation inhibitors that differ in their ribosomal binding site and kinetics but for which we lack any more detailed information, is a useful baseline reference for comparison, *i.e.*, an improved “null model”.

To clarify the purpose of our biophysical model and the reasons for not including more details, we revised the paragraphs in the Results part where we first explain the model (lines 109-116) and added a new paragraph in the Discussion (lines 472-487).

The most important issue 2: translational bottleneck model. The authors attempt to use growth-law model to explain a part of drug interactions (57%). Translational bottleneck model doesn't do much further (71%). On one hand, the author should link those two theories (I don't see the relation here). Do those artificial translational perturbation systems also observe the growth-law model? On the other hand, there are many uncertain things regarding using artificial translation perturbations to mimic the antibiotics. Since as the authors also point out, artificial titration of certain translational proteins is certainly a more pure way while a lot of antibiotics have secondary effects which make the situation be more complicated. Therefore, with the imperfectness of both the two models and those uncertainties in the model, the conclusion of the paper is not that convincing.

Relation of growth-law model and translation bottlenecks approach: We agree that it was not clear enough that the growth-law model and translation bottlenecks are two independent approaches to tackle the same problem. There may also be a misunderstanding here: The translation bottleneck approach is not a mathematical model – here, we merely convert growth rate measurements in two-dimensional antibiotic-inducer concentration gradients into predicted response surfaces for the combination of a real antibiotic with an idealized antibiotic that is mimicked by depleting the respective translation factor (see *e.g.*, Fig. 4c, 5a and Supplementary text, section “Remapping”). The only connection to the growth-law model is that, for this conversion, we need to assume a specific shape of the dose-response curve of the mimicked antibiotic, for which we simply use the shape predicted by the growth-law model (for a single antibiotic). We revised the explanation of the translation bottleneck approach in the Results part and now explain more clearly that it is independent from the biophysical model (lines 274-281); in particular, we explicitly state that the predictions of the translation bottleneck approach are not based on theory, but rather on empirical observations.

Fraction of drug interactions explained: It is correct that, applying stringent criteria, the percentage of explained drug interactions only increases moderately (from 57% to 71%) by the translation bottleneck approach. However, the additional interactions that are explained include several of the strongest interactions in our dataset and, in particular, the most striking suppressive interactions which were entirely unexplained by the biophysical model. Here, it is good to keep in mind that many relatively weak or additive interactions are included in the 57% explained by the biophysical model. Hence, these fractions do not reflect the “importance” of the drug interactions explained – the identification of a novel underlying mechanism of a drug interaction can be interesting even if it applies to relatively few drug interactions in our data set (*e.g.*, if there were only a single instance of strong synergy in a large drug interaction network, that would certainly still be interesting to understand).

Also, note that, due to the stringent criteria we apply, the 71% of explained drug interactions is a very conservative estimate. For a prediction to count as correct, we require that the vast majority of growth rates across the entire two-dimensional drug space closely matches the prediction (see “Quantitative

comparison of predicted and measured response surfaces”, lines 275-306 in Supplementary Methods) – this is not just a discrete decision about qualitative matches in drug interaction type. In particular, in the case of a suppressive interaction, the direction and magnitude of suppression (*i.e.*, which drug suppresses the other and how strongly) needs to be correctly predicted for it to be classified as a match. For comparison, we performed consistency checks (as described in the Supplementary Methods, lines 307-311) in which we pretend that one of the experimental replicates is a prediction of the other. Applying the same criteria as for our real predictions, this results in 75% of replicates being good predictors of one another. This indicates that the fraction of 71% faithfully predicted interactions is actually close to the maximum that can be achieved at our measurement precision. We have revised the Discussion (lines 418-434) to clarify these points.

Growth laws for artificial translational perturbations: Indeed, there is evidence that genetic translational perturbations similar to those we used in this work lead to the same negative correlation between ribosome content and growth rate as observed for chloramphenicol (and other translation inhibitors, see above). For example, when translation is slowed by mutations in *rpsL*, the same growth law is observed [Scott, M., *et al. Science* **330** (2010)]. Similarly, as mentioned above, lowering the expression of initiation factors agrees with the growth law [Cole, J.R., *et al. J. Mol. Bio.* **198** (1987); Olsson, C.L., *et al. Mol. Gen. Genet.* **250** (1996)]. It would be interesting to investigate this point for all translation factors, but this would require a massive effort, which is beyond the scope of our study. Importantly, the conclusions about drug interactions we draw using these translational perturbations (Figure 4 and 5) are independent of whether or not they obey the growth law. We have added an explanation of this point in the Results part where we introduce the translation bottleneck approach (lines 274-281) and where we explain the main conclusions from this approach.

Secondary effects of antibiotics: We certainly agree with the reviewer that antibiotics can have secondary effects that may cause some of the observed drug interactions. The point of the translation bottleneck approach is to test if the specific primary effect on translation of the antibiotic alone can explain the observed drug interactions. Thus, the purer way of perturbing translation by titrating translation factors enables us to disentangle drug interactions that are explained by the primary mode of action of the drugs from those that are not. In the latter case, drug interactions are likely caused by secondary effects of the drugs; these drug interactions remain unexplained by our approach, may have complex origins, and require a case-by-case investigation. However, our results show that the majority of drug interactions between ribosome inhibitors are explained by the primary action of these antibiotics on translation and the general physiological response of the cell to such perturbations. We consider this a major advance in our understanding of drug interaction mechanisms and the ability to mimic drug interactions genetically a key strength of our approach. To make this point clearer, we have revised the main text explaining the rationale for mimicking antibiotic effects with translation bottlenecks (lines 181-185, 238-259) and expanded the part about secondary effects and other drug interaction mechanisms in the Discussion (lines 472-493).

Minor comments:

To my knowledge, E. coli K-12 strain is not sensitive to fusidic acid due to the poor permeability (please clarify this issue). Therefore, I am not sure how the authors perform fusidic acid experiments.

The reviewer is correct that high concentrations of fusidic acid are needed to inhibit growth of *E. coli* K-12. This is due to poor membrane permeability [see Bennet, P.M., & Maaløe, O. *J. Mol. Bio.* **90** (1974)] and renders fusidic acid not useful in the clinic. Yet, at higher doses, fusidic acid does inhibit growth (minimal inhibitory concentration ~160 ug/mL, which is well below the solubility limit of 50 mg/mL) and

has been used in other *in vitro* studies [e.g., Yeh, P., *et al*, *Nat. Genet.* **38** (2006)]. We now mention this in the Supplementary Information (lines 81-84).

Moreover, for the growth rate measurement, I have a hard time to find an exact concentration of drug that used. Most data is an ambiguous IC50. This is unclear. I suggest the author to show a dose-dependence growth rate data for each drug with (exact concentrations of drug, instead of merely IC50), to standardize the result.

We added a panel in Supplementary Fig. 1, which shows the dose-response curves with absolute concentrations. However, we decided to keep all other plots in units of the IC50 as this facilitates the direct comparison of different drugs on a single scale.

The method of growth rate measurement is very weird to a microbial physiologist. Why does the author adopt such a method? The conventional OD measurement is very easy; you can simply do it with a micro-plate reader.

The reviewer is certainly correct that growth rate can be determined by optical density (OD) measurements; we have experience with that [e.g., Chevereau, G., & Bollenbach, T. *Mol. Syst. Biol.* **11** (2015)]. Here, we used a previously established luminescence assay [Kishony, R., & Leibler, S. *J. Biol.* **2** (2003); Chait, R., Craney, A., & Kishony, R. *Nature* **446** (2007); Yeh, P., *et al*, *Nat. Genet.* **38** (2006)], which offers higher sensitivity and precision than OD measurements and facilitated performing our measurements in high-throughput (~1,400 growth curves per experiment). This technique is based on the proportionality between luminescence intensity and number of cells [Kishony, R., & Leibler, S. *J. Biol.* **2** (2003)]. The technique allows detection of population growth over more than five orders of magnitude, which translates into the ability of tracking exponential increases in population size over more than 16 generations. We added more explanation of this technique in the Supplementary Information text (lines 19-28, 105-107) and in the new Supplementary Fig. 7. This figure shows control experiments confirming that growth rates measured by OD and by luminescence agree. See also our reply to the comments of reviewer #2 above.

REVIEWERS' COMMENTS:

Reviewer #2 (Remarks to the Author):

The revised manuscript has improved compared to the original submission. Nevertheless, I believe its interest for a general audience is fairly limited, which makes me recommend publishing it in a more specialized journal, e.g. Antimicrobial Agents and Chemotherapy.

Reviewer #3 (Remarks to the Author):

I have reviewed the previous version of this manuscript. The revised manuscript has been substantially improved. Specially, now with more thorough clarifications of the models in the rebuttal letter and manuscript, I have to admit that the major part of my previous criticisms is out of the scope of the current manuscript.

Anyway, I am impressed by the simplicity and predictive power of the model and the translation perturbation systems. I recommend its publication in Nature Communications.